# Dynamic structure of locomotor behavior in walking fruit flies

Alexander Y Katsov[1]*[†‡], Limor Freifeld[2,3§], Mark Horowitz[2], Seppe Kuehn[4,5,6], Thomas R Clandinin[1]

[1]Department of Neurobiology, Stanford University, Stanford, United States; [2]Department of Electrical Engineering, Stanford University, Stanford, United States; [3]Research Laboratory of Electronics, MIT Electrical Engineering and Computer Science Department, Cambridge, United States; [4]Center for the Physics of Living Cells, University of Illinois at Urbana-Champaign, Urbana, United States; [5]Center for Biophysics and Quantitative Biology, University of Illinois at Urbana-Champaign, Urbana, United States; [6]Department of Physics, University of Illinois at Urbana-Champaign, Urbana, United States

*For correspondence: akatsov@ rockefeller.edu

Present address: [†]Laboratory of Neural Circuits and Behavior, The Rockefeller University, New York, United States; [‡]Laboratory of Living Matter, The Rockefeller University, New York, United States; [§]Department of Neurobiology, Tel Aviv University, Tel Aviv, Israel

Competing interests: The authors declare that no competing interests exist.

**Abstract** The function of the brain is unlikely to be understood without an accurate description of its output, yet the nature of movement elements and their organization remains an open problem. Here, movement elements are identified from dynamics of walking in flies, using unbiased criteria. On one time scale, dynamics of walking are consistent over hundreds of milliseconds, allowing elementary features to be defined. Over longer periods, walking is well described by a stochastic process composed of these elementary features, and a generative model of this process reproduces individual behavior sequences accurately over seconds or longer. Within elementary features, velocities diverge, suggesting that dynamical stability of movement elements is a weak behavioral constraint. Rather, long-term instability can be limited by the finite memory between these elementary features. This structure suggests how complex dynamics may arise in biological systems from elements whose combination need not be tuned for dynamic stability.

## Introduction

Behaving animals can act as quickly as their nervous systems allow, as slowly as their environments fluctuate, or over intervals determined by some task. For instance, behaviors may change quickly to avoid predators, slowly to adjust to seasons, or at the intervals of locomotor steps, song phrases, or nest-building stages. How different behaviors between these limits fit together remains poorly understood.

Classical ethological studies examined different stereotyped, goal-driven behaviors including feeding, courtship rituals, aggressive encounters, and escape sequences (*Tinbergen, 1963*). To human observers, many of these behaviors consisted of recognizable movements that often occurred in characteristic sequences. These movements were inferred to be the fundamental units of behavior, and their modularity suggested a similarly modular organization in neural control (*Tinbergen, 1950*; *Simmons and Young, 2010*). Yet on the whole, animals can adjust their movements in both discrete and graded ways. For instance, houseflies track other flies in flight using a combination of stereotyped saccadic course corrections and smooth pursuit (*Wagner, 1986*), and primates track small targets using both saccadic and smooth orienting movements (*Fuchs, 1967*). While discrete movements can be imprecise, gradual velocity adjustments can be used to tune movements quickly with fine resolution. However, graded movement control, as a dynamical system, requires stability to

be useful. As stable implementation of arbitrary dynamical systems is challenging (*Smale, 1966*), graded control may impose limits on movement complexity or risk susceptibility to unpredictable behavior over time. Thus, while an intuitive description of movement has converged on modularity in both high and low level motor control (reviewed in *Giszter and Hart, 2013*; *Flash and Hochner, 2005*), variability and flexibility of ethological units have long been appreciated (*Barlow, 1968*), and task-specific coordination of motor units down to individual muscles has been suggested (*Kutch et al., 2008*; *Valero-Cuevas et al., 2009*). It remains unclear whether degrees of freedom in movements are themselves limited by a finite set of functional modules, or may be adjusted continuously and in a task-specific manner (*Tresch and Jarc, 2009*). In invertebrates, it has been argued that behavior modules represent about half of all behaviors (*Berman et al., 2014*), but it has also been argued that apparent modules represent mere extremes within a continuum of behaviors (*Gallagher et al., 2013*; *Szigeti et al., 2015*; *Hums et al., 2016*). Thus, the extent to which behavior is modular remains unresolved.

Recent advances in monitoring behavior, computing power, and statistical tools have encouraged efforts to examine behavioral organization in an unbiased manner. However, there is scant consensus on methods or criteria for unbiased segmentation. A variety of dimensionality reduction techniques using both linear and non-linear approaches have been applied to dissecting limb and animal movements in both vertebrates and invertebrates (*Fod et al., 2002*; *Del Vecchio et al., 2003*, *Avella and Bizzi, 2005*; *Stephens et al., 2008*; *2010*, *Braun et al., 2010*; *Gallagher et al., 2013*; *Mendes et al., 2013*; *Berman et al., 2014*; *Vogelstein et al., 2014*; *Schwarz et al., 2015*; *Wiltschko et al., 2015*). In addition, a number of approaches have used human observations to train statistical models to identify behavioral patterns. For example, the movements of both flies and worms have been described as sequences of behavioral events that were originally selected by human observers (*Croll, 1975*; *Pierce-Shimomura et al., 1999*; *Branson et al., 2009*; *Dankert et al., 2009*; *Kabra et al., 2013*; *Kain et al., 2013*). These methods have allowed human-observed behaviors to be scored more efficiently, enabling high-throughput quantification of gender and individual differences, or encounter types between individuals (*Branson et al., 2009*; *Dankert et al., 2009*). However, given the overall diversity of approaches that have been applied, and the corresponding differences in their conclusions, identifying principles of behavioral organization across different behaviors and organisms remains challenging. Here we develop a set of criteria for behavior segmentation that may be applied wherever time series measurements of behavior can be obtained.

As a simple model of spontaneous behavior, we studied fruit fly locomotion. Due to the fly's relatively simple nervous systems and compact behavioral repertoire, this model system provides an opportunity for high-throughput studies of behavior that can be linked with circuit function. In this model system, a variety of behavioral sequences have been studied (*Strauss and Heisenberg, 1990*; *Pick and Strauss, 2005*; *Branson et al., 2009*; *Chen et al., 2002*; *Spieth, 1974*), and powerful genetic tools can be used to link behavior to circuits (*Simpson, 2009*).

To build a catalog of spontaneous behaviors, we acquired large datasets capturing the locomotor movements of freely walking animals. We then identified patterns in movement dynamics from body velocity time series using an iterative ICA procedure, and defined unique or interchangeable behavior components from their occurrence statistics. We found that walking behavior can be decomposed into a small number of patterns that occur largely independently of all but the immediately preceding behavior. Next, we developed a statistical model that captured sequences of these patterns over longer timescales, and tested the ability of this model to generate sets of synthetic behavior trajectories matching real fly behaviors. On short time scales, we find a continuum of behaviors whose variation can be captured by a small number of parameters. Over longer times, variation grows, but behavior is predictable because it resolves into a small number of distinct, finite memory episodes, corresponding to elementary walking patterns. As a result, these studies systematically connect the time scales of velocity modulation on the order of the action-perception cycle with longer time scales at which ethological units of behavior are observed.

## Results

### Behavior shows consistency over short periods, describing the extent of behavior episodes

We set out to characterize walking at time scales from tens of milliseconds to a few seconds, beginning at the time scale of the fly action-perception cycle and extending to the time scale of locomotor behaviors (*David, 1984*; *Nagle and Bell, 1987*). Spontaneous walking behavior of female flies was examined in a uniformly illuminated environment without systematically varying visual, auditory or olfactory stimuli. We were interested in finding reproducible behaviors and sought to eliminate individual differences and behavioral changes that might emerge over extended periods of time. To minimize slow behavior changes due to fatigue, circadian phase, or age, well-fed 2–3 day-old flies (n = 7364) walking on a glass surface under dim illumination were tracked alone or in groups, in 10 minute trials during 2 hours of peak circadian activity. In aggregate, our datasets comprised over 1000 fly-hours of behavior, and contained over $10^6$ trajectories (*Katsov et al., 2017*). Three datasets were collected for different purposes: a dataset of $\sim 10^6$ trajectories that densely sampled walking behavior on short time scales (Dataset S, trajectory length *mean* = 1.5 sec, $P_{95}$ = [0.22,5.1] sec), a dataset of $\sim 10^4$ longer trajectories in a large arena (Dataset L, *mean* = 5.22 sec, $P_{95}$ = [0.42,14.55] sec), and a dataset of $\sim 10^4$ trajectories from isolated individuals (Dataset A, *mean* = 2.2 sec, $P_{95}$ = [0.63,7.2] sec). Given these large datasets, we could effectively sample a wide range of spontaneous locomotor behaviors.

Locomotor behaviors are commonly defined using simple speed heuristics. For instance, turns have been defined in several species, including flies, by setting a threshold on rotational speed, and runs and pauses have been defined by thresholds on translation (*Berg and Brown, 1972*; *Pierce-Shimomura et al., 1999*; *Geurten et al., 2010*; *Drai et al., 2000*; *Wolf et al., 2002*). However, behavior categories are not always reflected unambiguously in instantaneous speeds. For instance, speed distributions with multiple modes can reflect distinct behaviors, each represented by a typical speed, but can also reflect biomechanical or other constraints that suppress some speeds in all behaviors, producing troughs in the distribution. To use more information, we looked at movement properties over time.

As Drosophila shows little independent head movement in walking maneuvers (*Geurten et al., 2014*; *Fujiwara et al., 2017*), we measured displacement of the entire animal as an oblong shape on a two dimensional surface and inferred the position of the head statistically with greater than 99.8% accuracy (*Figure 1A*; *Katsov and Clandinin, 2008*). Three velocity components, translation, rotation, and side-slip, $\vec{v} = (v_T, v_R, v_S)$, completely describe walking at this level (*Figure 1B*). These velocity components, plus accelerations $\dot{\vec{v}} = (\dot{v}_T, \dot{v}_R, \dot{v}_S)$ were measured from $10^6$ individual trajectories (*Figure 1C*).

Movement properties were first examined over a small time step, 33 milliseconds, in a velocity-acceleration phase space. In this space, starting from velocity trajectories, the average acceleration at each $\vec{v}$, $\mathbf{E}\left[\dot{\vec{v}} \mid \vec{v}\right]$, was used to construct a kinematic field (Materials and methods). This field shows average tendencies at different velocities. For example, when flies are not turning (*Figure 1D*, $v_R \approx 0$), but have high side-slip they tend to speed up (arrows up), while at low side-slip they tend to slow down (center of panel, arrows down). Altogether, this field constitutes a local description of behavior dynamics as it explicitly describes average dynamics at each velocity bin. For locomotor behavior, this description captures the mapping between observed behavior and aggregate forces due to neural control and biomechanical constraints, as indirectly represented in the accelerations observed at each velocity. If locomotor behaviors are separated by distinct combinations of velocity-acceleration pairs, different regions of this space should describe distinct behaviors well. However, in the kinematic field, mean dispersion about the average accelerations is greater than five times the local averages, many-fold greater than our measurement error (Materials and methods). This large dispersion may represent intrinsic behavior variability, or suggest that distinct behaviors are not well represented in their average tendencies at one time scale.

We considered three possible explanations for this large dispersion (*Figure 2*). First, though unlikely, the maneuvers of walking flies, like those of flying insects, may be relatively unstable (*Taylor and Thomas, 2003*; *Sun and Xiong, 2005*). As a consequence, two flies initiating a turn in the same way, with nearly identical velocity, might move in very different ways a short time later. A

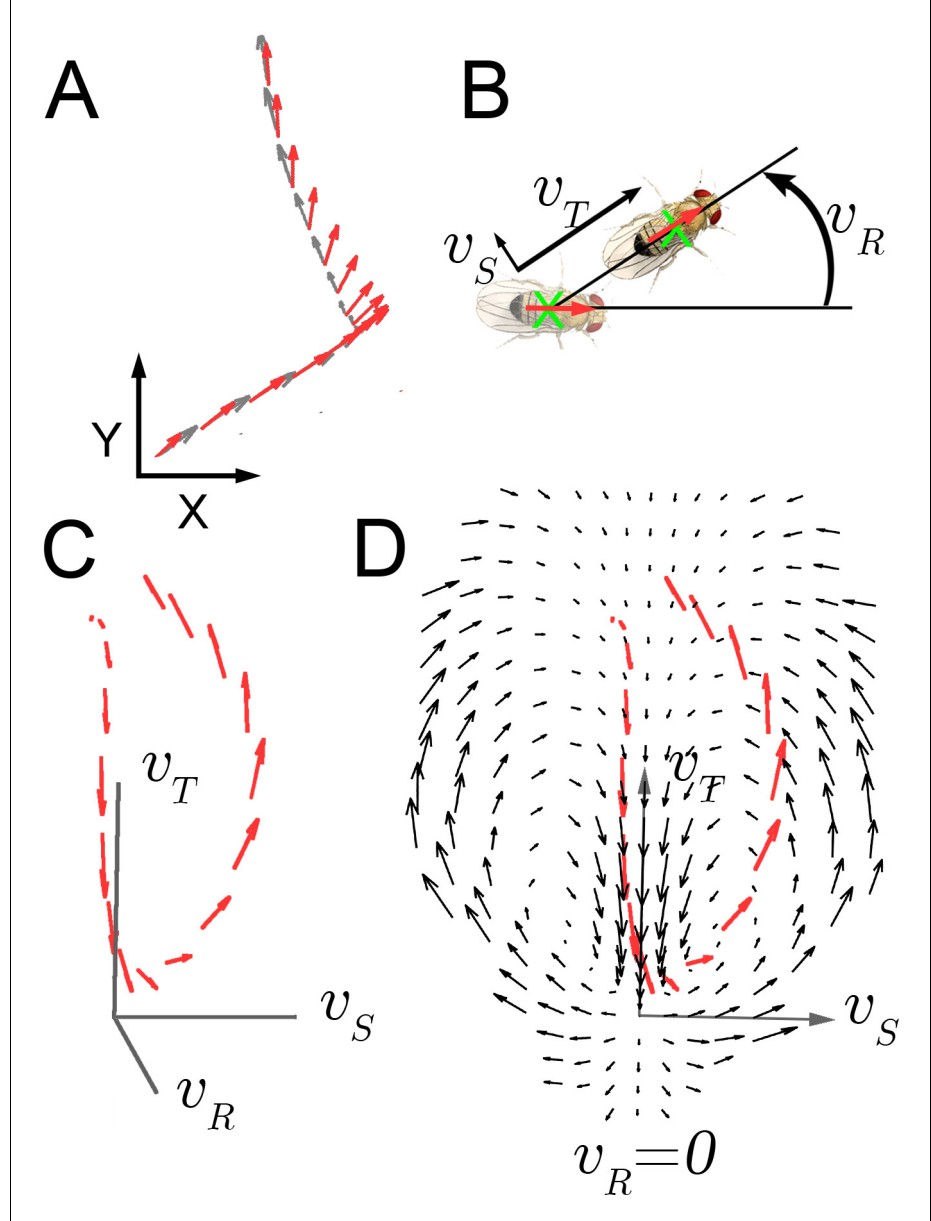

**Figure 1.** Velocity-acceleration phase space captures aggregate constraints on behavior. (**A**) An example trajectory fragment of an individual fly walking on a surface. The animal's orientation (red arrows) and displacement (gray arrows) are tracked at 30 frames per second. (**B**) A schematic illustration of the three velocity components $\vec{v} = (v_T, v_R, v_S)$ around the major body axis of a fly. These three components completely describe whole-animal movement in two dimensions. (**C**) Example trajectory in velocity phase space parameterized by $\vec{v} = (v_T, v_R, v_S)$. Red arrows denote the magnitude of the acceleration vector at each velocity. (**D**) A single plane in the 3-dimensional phase space, $\mathbf{E}\left[\dot{\vec{v}} \mid \vec{v}\right]$, $\vec{v} = (v_T, v_R = 0, v_S)$ (black arrows), with an example trajectory (red arrows), projected into this plane (x-axis: $v_S$; y-axis: $v_T$).

broad range of accelerations at any given velocity will result, and behavior dynamics, represented by accelerations, would appear poorly constrained. This possibility predicts that if one were to follow trajectories in velocity space, behaviors would diverge quickly from initially similar velocities (*Figure 2A*). Second, behavior dynamics may be stable, but related trajectories can be reasonably expected to vary (*Barlow, 1968*). Hence, even if distinct behaviors were well separated in velocity phase space, a range of accelerations at any given velocity would still be observed. However, unlike

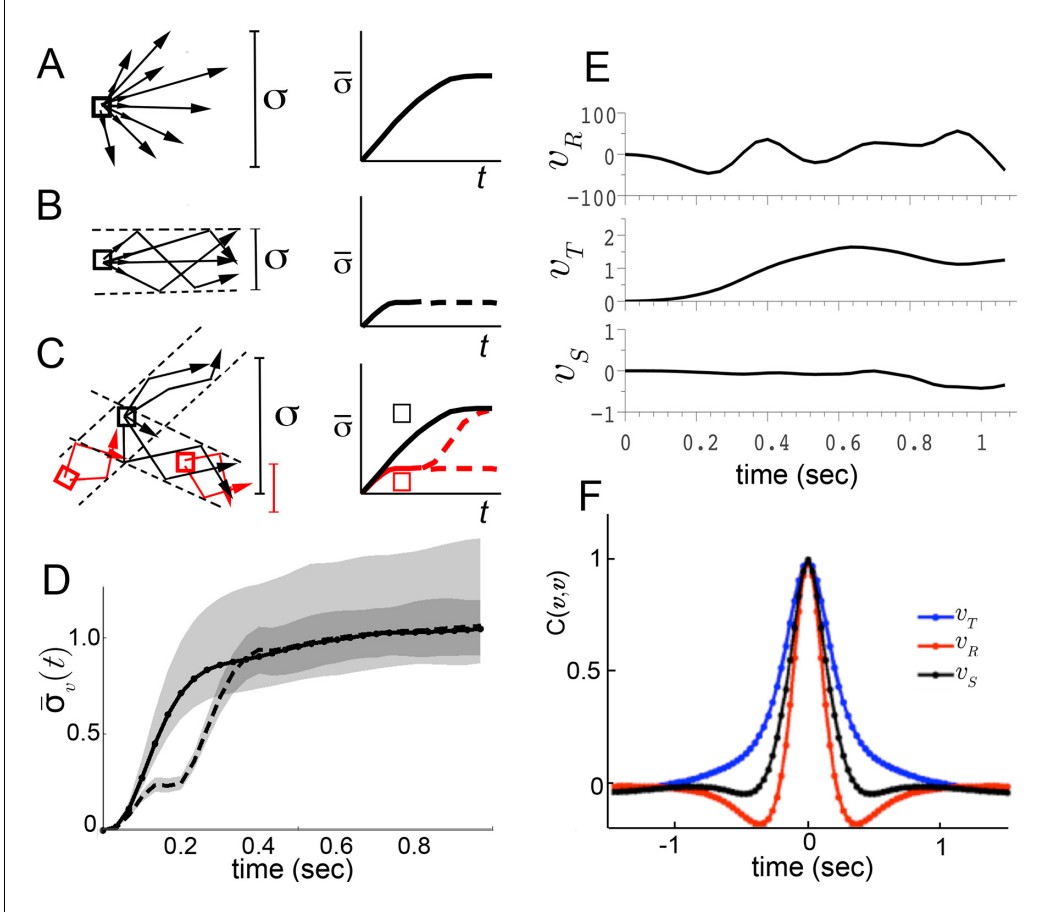

**Figure 2.** Trajectory divergence in velocity phase space and definition of a behavior episode around $v_R$ extrema. (A) Starting from a small neighborhood of similar velocities (box), diverging trajectories spread throughout velocity phase space, increasing in a measure of their spread. (B) Starting from the same neighborhood, divergence of trajectories corresponding to modal patterns is bounded for the duration of each pattern. (C) Bounded or unbounded divergence can be seen when modal patterns overlap in velocity phase space, depending on when trajectories are sampled in the course of these patterns. (D) The normalized standard deviation of trajectories at time $t$ after passage through a small neighborhood, averaged over a random sample of $N_B = 200$ neighborhoods, $\sigma(t) = \frac{1}{\sigma[v_T]\sigma[v_R]\sigma[v_S]} \langle \sigma_t[v_T]\sigma_t[v_R]\sigma_t[v_S] \rangle_{N_B}$ (*Equation 1*, Materials and methods). Shaded areas represent 95% confidence intervals of the estimate of $\sigma(t)$. Neighborhoods were sampled from anywhere in velocity space, and for each neighborhood all trajectories were included (dot-solid line), or only those with a turn peak 100 ms after passing the neighborhood (dashed line). (E) An example trajectory fragment depicting $v_R(\tau)$ [° s$^{-1}$], $v_T(\tau)$ and $v_S(\tau)$ [cm s$^{-1}$]. (F) The autocorrelation functions of $v_R(\tau)$, $v_T(\tau)$ and $v_S(\tau)$.

the first possibility, the second predicts that movements of one type will maintain velocities that are more similar to each other over time than to other movement types (*Figure 2B*). Related behaviors should remain close in velocity space over time, in distinct regions from other behaviors. The final possibility is that distinct movement patterns use distinct accelerations at similar velocities, and therefore overlap in velocity phase space. As a result, different patterns contribute different accelerations to the same velocity neighborhood. In this case, initially similar velocities would remain similar when behavior is committed to a particular movement type, but will diverge at other times (*Figure 2C*, red, black squares, respectively).

To test these alternatives, we randomly sampled small velocity neighborhoods, each corresponding to approximately 0.004% of velocity space, and measured divergence of trajectories from each neighborhood over time (Materials and methods). On average, trajectories quickly diverged up to the extent of the entire velocity space (*Figure 2D*, solid line), excluding the possibility that stable behaviors are well-separated in instantaneous velocities (*Figure 2B*). Of the remaining possibilities, is walking behavior unstable (*Figure 2A*), or composed of distinct patterns that re-use similar velocities (*Figure 2C*)? We reasoned that if stable behaviors exist, they should be most consistent when

behavior is committed to a particular movement type. Therefore, we looked for behavior episodes between locomotor adjustments. Whenever locomotor behavior is adjusted, some force must be applied to alter velocity. As a result, behavior adjustments will correspond to accelerations or decelerations. As flies accelerate or decelerate for finite periods of time, episodes of behavior adjustment must have extrema. Acceleration extrema correspond to inflections in velocity and hence we chose to examine trajectory intervals that included at least two velocity inflections, bounding an interval between behavior adjustments. We observed that the rotational component fluctuated throughout all trajectories, and displayed a steep autocorrelation falloff (*Figure 2E,F*). Indeed, local peaks in $v_R$ arise frequently enough, with an inter-peak interval of $250 \pm 110$ ms, that almost every trajectory can be represented in its entirety by short trajectory fragments surrounding $v_R$ peaks. When fragments include most of each interval before and after a peak, they are nearly guaranteed to include behavior adjustments on either side of the peak, capturing a relevant behavior interval. To test whether behavior divergence is ever bounded during these episodes, we examined behavior starting from defined times preceding velocity peaks. Specifically, we selected velocity neighborhoods randomly throughout velocity space again, but only examined divergence of trajectories where a $v_R$ peak occurred a defined time later, ranging from 70 ms to 170 ms. Divergence should be unaffected by this selection if behavior is unstable (*Figure 2A*). To the contrary, we observed that on average divergence plateaued for up to 200 ms when trajectories were sampled up to 130 ms before $v_R$ peaks, but not earlier (*Figure 2D*, dashed line, and d.n.s). This period of bounded divergence suggested that there exist behavior patterns that remain consistent for about 200 ms, beginning from approximately the middle of the average interpeak interval. Taken together, our findings excluded possibilities A and B, suggesting that different movements must overlap in velocity phase space (*Figure 2C*), while consistent behaviors may be defined with respect to $v_R$ peaks.

## Unbiased segmentation of trajectories defines modal movement patterns

We next sought to separate behavior patterns statistically. In walking flies, certain movement patterns stand out for human observers: for instance, stops, crabwalks, sharp turns, straight and curved walks (eg: *Strauss and Heisenberg, 1990*; *Branson et al., 2009*; *Kain et al., 2013*; *Geurten et al., 2014*). However, on closer inspection, even seemingly stereotyped patterns like sharp turns present a variety of movements, producing a range of very large to barely perceptible orientation changes. Does this range include distinct movement patterns? Conversely, are movement patterns that look similar to human observers really used by animals as if they are the same? Without explicit human guidance, approaches to behavior segmentation have been based on models of joint kinematics, movement dynamics, or muscle synergies (eg: *Fod et al., 2002*; *Del Vecchio et al., 2003*; *d'Avella and Bizzi, 2005*), or on classification of movements from distributions of instantaneous velocities (*Braun et al., 2010*; *Gallagher et al., 2013*), or postures (*Stephens et al., 2008*, *2010*; *Berman et al., 2014*; *Schwarz et al., 2015*; *Wiltschko et al., 2015*), or experimental activation of neuron groups (*Vogelstein et al., 2014*). Our strategy was to identify any behaviors that may be different in dynamic structure, and then to use statistical relationships between these behaviors to determine if multiple behaviors belong in the same group, or whether one identified behavior is used in different ways so as to suggest multiple underlying states.

Distinct behaviors arise from differences in how flies accelerate or decelerate. Due to inertia, velocities cannot change instantaneously. However, as observed in *Figure 2D*, velocities may or may not remain similar over time, depending on what phase of behavior is examined. Because behaviors maintained similarity around $v_R$ peaks, trajectory fragments surrounding $v_R$ peaks were chosen for analysis. Specifically, fragments were selected including 550 ms before and after each $v_R$ peak, encompassing over 99% of all interpeak intervals (*Figure 3A*, *Box 1*). This fragment length extended beyond the average period of bounded divergence to include as many full intervals between behavior adjustments as possible, allowing for extended behaviors. As a result, a series of partially overlapping fragments were extracted from each trajectory. These fragments were aligned in time to when a peak occurred in $v_R$, thereby aligning the adjoining rising and falling $v_R$ signals across trajectories. In this aligned dataset, distinct movement patterns would be marked by different magnitudes of $v_R$ peaks, different temporal patterns of changes in velocities around turn peaks, or both. The aligned dataset was represented in a matrix, where each column contained a different fragment around a unique $v_R$ peak, and each row represented a step in time relative to that peak, for each velocity

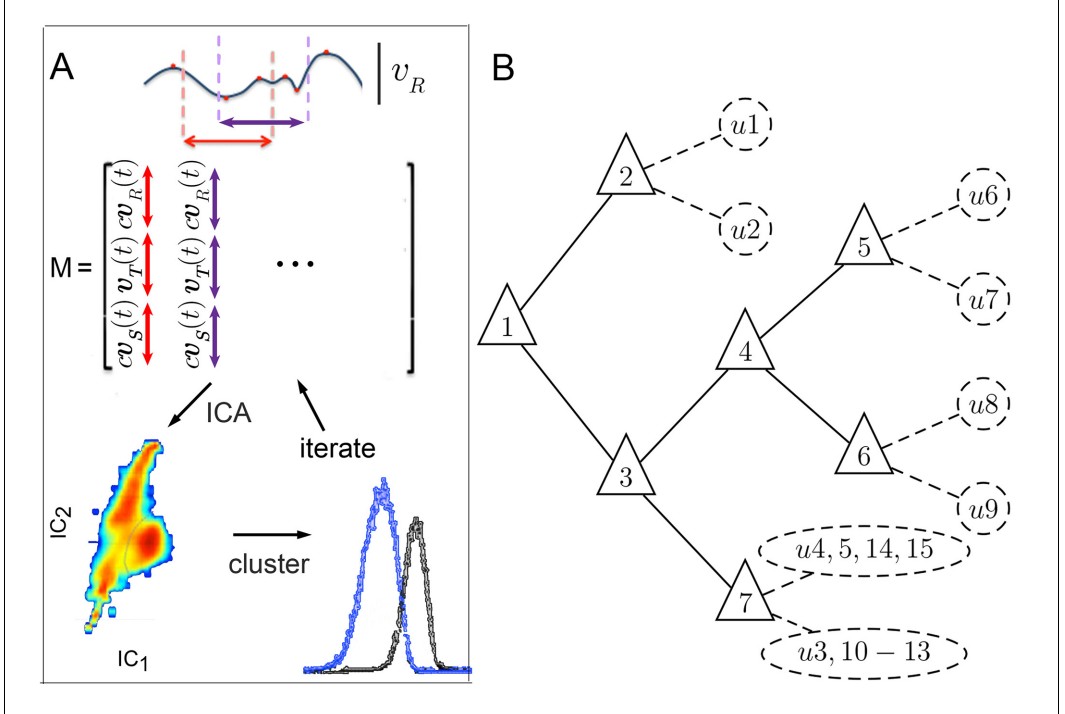

**Figure 3.** Iterative ICA segmentation reveals dynamic submodes. (**A**) Iterative ICA procedure. Local $v_R$ peaks are identified in a single trajectory (red dots). Fragments of trajectories are then selected around $v_R$ (dashed lines), and all velocity components are aligned in time on $v_R$ extrema in matrix **M**. In **M**, turn direction is normalized while preserving the relative direction of side slip by multiplying $v_R$ and $v_S$ by $c$, the sign of $v_R$ extremum at aligned time 0. **M** dimensionality is then reduced and independent components are identified. In these independent components, velocity trajectories are represented by a coefficient in each component. If a multimodal coefficient distribution is found, coefficient clusters corresponding to distinct trajectory fragment subsets are separated, and the entire procedure is repeated iteratively on each fragment subset to identify distinct submodes. (**B**) Segmentation tree showing decision points (triangles), and submodes (dashed circles). 15 submodes were identified, but the decision tree is drawn up to branching levels supported by Markov modeling (*Figure 5*, *Figure 5—figure supplement 1*). Submodes are described in *Figure 4* and *Figure 4—figure supplement 1*.

component. Different columns then contained either variants of related behaviors, or distinct movement patterns.

To identify potentially different movement patterns, we looked for simple descriptors of velocity fragments, asking which fragments are well captured by the same set of descriptors, and which fragments require different sets of descriptors. These descriptors were inferred from patterns of variation in the fragment set, using independent components analysis, ICA (*Hyvärinen, 1999*; *Hyvärinen and Oja, 2000*; *Himberg et al., 2004*), *Box 1, 2*, Materials and methods). Using ICA, the shapes of trajectory fragments in velocity space were reduced to a set of independent parameters. These parameters may be thought of as degrees of freedom in the temporal structure of velocity trajectories. An individual trajectory fragment, as one trajectory shape described by a set of parameters, is represented by a coefficient in each parameter. To separate potentially unrelated fragments, we assumed that under most circumstances, parameter coefficients should be distributed unimodally if a parameter represents one degree of freedom of a single movement pattern. Alternatively, a multimodal coefficient distribution indicates that movement represented in this parameter may be impeded by some mechanical constraint, that movement is not well represented in this parameter, or that more than one movement type is present. For these reasons, coefficient distributions were examined once independent components were identified in a fragment dataset, starting from the complete dataset consisting of $2.9 \cdot 10^6$ trajectory fragments. Whenever a coefficient distribution showed multiple modes or inflections, raising the possibility of contributions from multiple movement patterns, data were separated by mode and the smaller pool of trajectory fragments corresponding to each subset was reanalyzed to find its own independent components. Because different

## Box 1. Segmentation of movement patterns, set-up.

Starting with the full set of trajectories $\vec{v}(\tau) = [v_T, v_R, v_S](\tau)$, times $\{t_e\}$ of local extrema in $v_R(\tau)$ were identified and trajectory fragments were isolated around each extremum, $\vec{v}(\tau = t_e)$. These fragments, $\vec{v}(t')$, $t' = \tau - t_e$, were normalized to the sign of $v_R(t' = 0)$, making all turns at $t' = 0$ positive but preserving side-slip direction relative to the turn. Each fragment of $n$ time samples was represented as a $(3n \times 1)$ column vector:

$$\mathbf{v} = \begin{bmatrix} cv_R(t') \\ v_T(t') \\ cv_S(t') \end{bmatrix}, \; c = sgn[v_R(t' = 0)]$$

Then, vectors $\mathbf{v}^{(j)}, j = 1, \ldots, N$, representing $N$ $v_R$ peaks, were concatenated horizontally into a $(3n \times N)$ matrix $\mathbf{M}$, such that each row $i$ was a velocity $cv_R$, $v_T$ or $cv_S$ at the same time $t'$ from a $v_R$ peak at $t' = 0$;

$$\mathbf{M} = \begin{bmatrix} \mathbf{v}^{(1)} & \mathbf{v}^{(2)} & \cdots & \mathbf{v}^{(N)} \end{bmatrix}$$

Correlations between rows in this matrix relate velocity components at aligned times $t'$ from $v_R$ peaks in the dataset. Because velocity components differ in range and distributions, which can also change over time relative to $v_R$ peaks, rows were standardized to obtain matrix

$$\mathbf{Z} = \begin{bmatrix} \mathbf{z}^{(1)} & \mathbf{z}^{(2)} & \cdots & \mathbf{z}^{(N)} \end{bmatrix}, \; z_i^{(j)} = \frac{\mathbf{v}_i^{(j)} - \langle \mathbf{v}_i^{(j)} \rangle_j}{\sigma[\mathbf{v}_i]}$$

This matrix may contain two kinds of mixtures of movement patterns. First, each column (or a part of it in time) may represent a variant of a single movement pattern. This movement pattern may be described in terms of independently adjustable parts of its temporal structure, or independent components. Framed in terms of the standard ICA generative model (**Hyvärinen and Oja, 2000**), it is hypothesized that vectors of this matrix can be described as:

$$\mathbf{z}^{(j)} = \mathbf{A}\mathbf{s}^{(j)}, \text{ or } \mathbf{z}^{(j)} = \sum_i^D \mathbf{a}_i s_{ij}$$

where $\mathbf{a}_i$ are columns of a mixing matrix $\mathbf{A}$, and $s_{ij}$ is a coefficient of sample $j$ in the $i^{th}$ basis vector, of the $D$ basis vectors that span the space of independent components of $\mathbf{Z}$. This basis vector set represents degrees of freedom in velocity trajectory shapes over time. $s_{ij}$ may take on continuous or discrete values, corresponding to continuous or discrete modes of movement control. The matrix $\mathbf{A}$ produces time-evolving trajectories by giving each scalar $s_{ij}$ an extent in time, and mixing these components. If movement dynamics can be represented by fewer degrees of freedom than dimensions of the standardized velocity-time vector $\mathbf{z}$, then $D < |\mathbf{z}^{(j)}| = 3n$.
The second kind of mixture in $\mathbf{Z}$ may be a mixture of distinct movement patterns, such that any given column (or part of it in time) contains one type of movement pattern, while different columns may contain qualitatively different patterns. If all movement dynamics are subject to the same degrees of freedom, then the same mixing matrix $\mathbf{A}$ applies to the entire dataset, and the space of independent components $\{\mathbf{s}\}$ is spanned by the same basis vector set $\beta$. Alternatively, different mixing matrices $\mathbf{A}^{(C)}$ and independent components $\beta^{(C)}$ describe each subset $C$, corresponding to distinct movement patterns.

movement patterns could arise at different frequencies, and require disambiguation of different velocity components on distinct time scales, the entire procedure was repeated iteratively for each

subset of raw velocity trajectories corresponding to a distinct coefficient subset, until further iteration produced no separable data features, did not converge, or produced independent components that did not differ from previous iteration (*Box 2* and Materials and methods).

Walking patterns classified in this way were termed submodes, as this analysis favored splitting over lumping of distinct behaviors and left open the possibility that identified behaviors may be subsets of larger behavior groups. Submodes included stops and dithers, when flies shifted around while remaining in one place, straight runs, smooth turns and several kinds of sharp turns, as well as different crabwalks marked by sideways motion. In addition, several variations of a sharp turn were found that included sideways or backward movement at high velocity (*Figure 3B*, *Figure 4* and *Figure 4—figure supplement 1*). As no more than 5–6 parameters were needed to account for at least

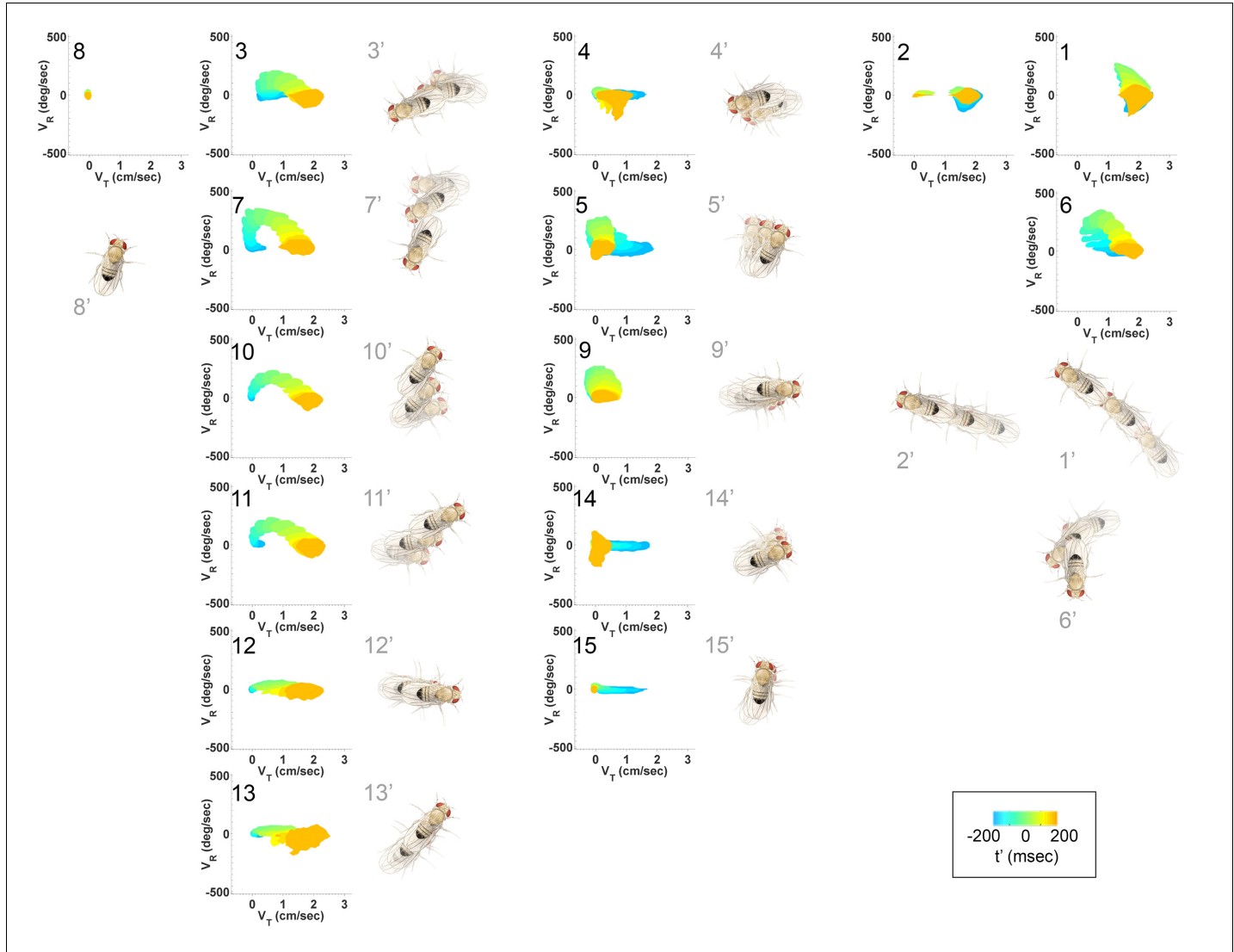

**Figure 4.** Submode velocity profiles for each submode. (1-15) Velocity distributions over time for each identified submode. The central 68% of the distribution is shown at each time point for up to 200 ms around peak $v_R$. The time within each submode relative turn peak is color-coded. Submodes are shown in columns, by membership in modes $I - V$ (*Figure 5*). (1'–15') Example trajectory fragment of each submode type is illustrated. Three time points are shown per trajectory, each spaced 100 ms apart. A diagrammed fly illustrates the centroid location and orientation of an animal at each time point. Note that only one time point illustrates submode 8 because this submode consists of stops.

The following figure supplement is available for figure 4:

**Figure supplement 1.** Example trajectory fragments from each submode class.

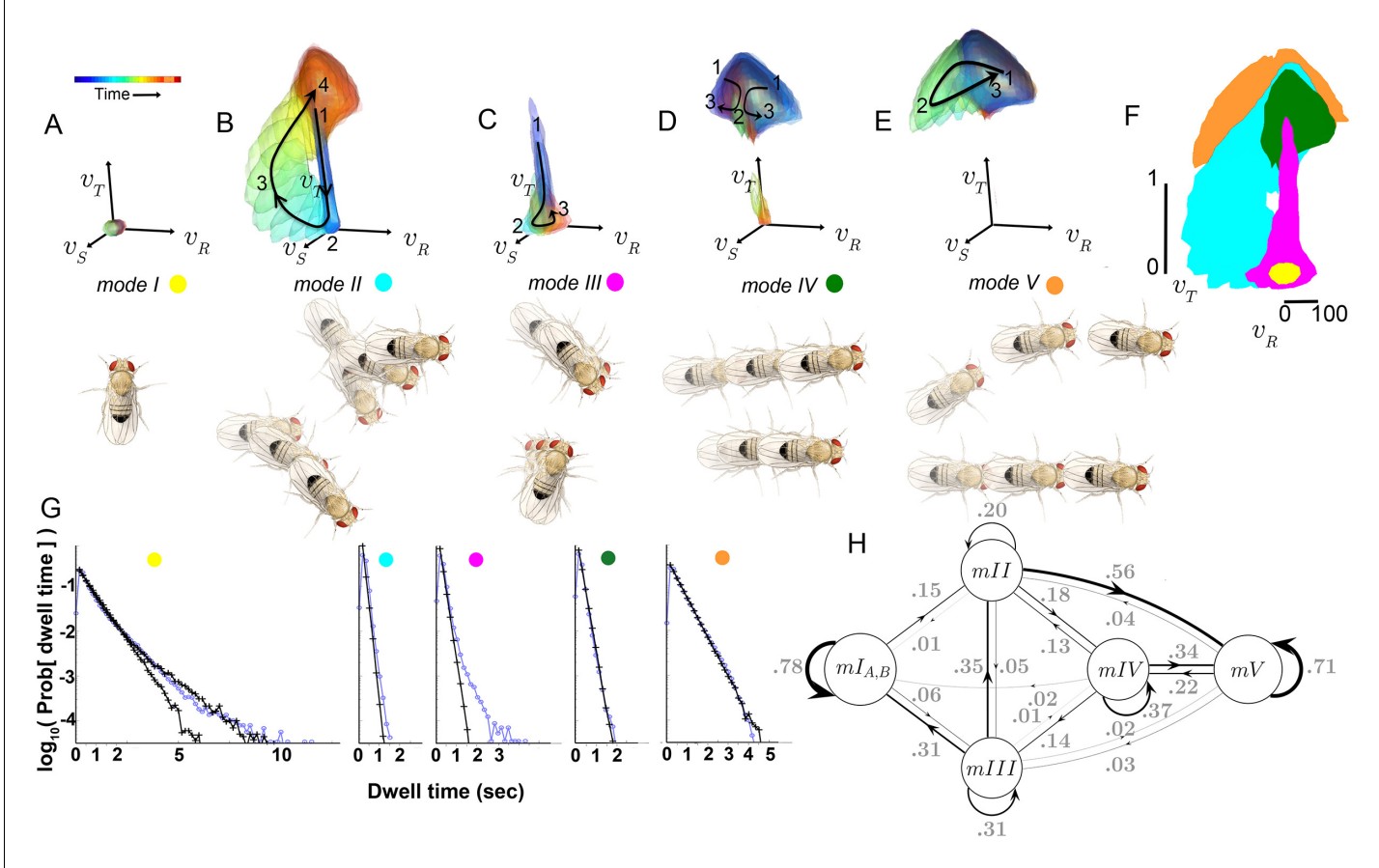

**Figure 5.** Identified movement patterns as distributions in velocity phase space. (A–E) Top: Modes are described as velocity distributions over time, termed 'velocity profiles.' The central 68% of the velocity distribution at each time point around peak $v_R$ is shown. The time relative to a turn peak within each mode is color-coded; green represents time of peak $v_R$. Bottom: Example trajectory fragments from each mode are illustrated. Except for *Mode I*, two trajectories are shown per mode, three time points per trajectory, each spaced 100 ms apart. A diagrammed fly illustrates the centroid location and orientation of an animal at each time point. Note that only one time point illustrates *Mode I* because this mode consists of stops. (F) Projection of all mode velocity profiles into a $v_T - v_R$ plane. Note that modes overlap in this joint velocity projection, and modes spanning smaller velocity regions are plotted on top of broader modes. (G) Residence time distributions, by mode, from real (light blue) and model-generated (black) trajectories. Two model-generated distributions are plotted for *mode I* (left panel): a Markov model without hidden states captures short residence times in the real distribution, while a Markov model that included an additional, non-communicating hidden state captures both short and long residence times. (H) The chosen 5-state Markov model, *MM5*. Edges are labeled with transition probabilities between modes. This model does not represent hidden substates of *mode I*.

The following figure supplements are available for figure 5:

**Figure supplement 1.** Grouping submodes by kinetic criteria to define a behavioral model.

**Figure supplement 2.** Information between modes in sequences.

85% of data variance at each iteration of the classification procedure, each movement pattern was reasonably well captured by a small number of parameters (see Materials and methods). Altogether, 15 submodes were identified that tiled individual trajectories from a dataset collected from thousands of individuals, likely accounting for all major walking patterns of female flies.

As a whole, the submode set included movements salient for human observers (*Branson et al., 2009*; *Kain et al., 2013*, pers. observation), but the extent and composition of some submodes and the diversity of others was not intuitive. For instance, turns represented in *submodes 7,10,* and *11* (*Figure 4*, *Figure 4—figure supplement 1*) were easy to pick out as one kind of behavior, but not necessarily three. Some human calls picked out features that were not identified as individual

## Box 2. Segmentation iteration.

(1) Construct matrix $\mathbf{M}$ from trajectory fragments around $v_R$ peaks. Include all velocity components (**Box 1**, and Materials and methods **Equation 3**). Columns are aligned in time with respect to $v_R$ peaks.

(2) Standardize, whiten rows of $\mathbf{M}$ to obtain matrices $\mathbf{Z}$ and $\mathbf{Z}^w$, respectively (Materials and methods **Equations 4–5**).

(3) Select the largest principal components (PCs) to retain $>85\%$ of total data variance. Eliminate rows corresponding to the remaining PCs from $\mathbf{Z}^w$, obtaining a reduced matrix $\widetilde{\mathbf{Z}}^w$.

(4) Apply Independent Components Analysis (ICA) to $\widetilde{\mathbf{Z}}^w$. Repeat with random initial guesses for ICs.

(5) Examine ICs and data projected into ICs

**if** *ICs fail to converge* **then**

 (5.1) Terminate procedure. Examine previous steps; reduce velocity-time points in $\mathbf{M}$ (dimensions of interest) as warranted. Proceed from step 1.

**else if** *ICs represent the same dimensions as pre-ICA* **then**

 (5.2) Terminate procedure. Trajectories in matrix $\mathbf{M}$ represent a submode

**else**

 (5.3) Project trajectories into ICs

 **if** *Parameter distribution in ICs contains no separable features* **then**

 (5.3.1) Terminate procedure. Trajectories in matrix $\mathbf{M}$ represent a submode

 **else**

 (5.3.2) Separate features in IC projections.

 (5.3.3) Identify trajectory fragments that correspond to each feature class in ICs.

 (5.3.4) Repeat, from Step 1, for each data subset.

 **end**

**end**

submodes: for instance, reversals did not appear in submodes on their own, but as part of more complex movement patterns. Similarly, *submode 1* spanned a range of movements from clear turns to relatively straight walks, while *submode 2* included both average and very slow walks, sometimes but not always with a clear side-slip component. Are such different movements really associated together? Conversely, five submodes represented more or less subtle variants of sharp turns (*submodes 3,6,7,10,11*) and several submodes included various subtle rotations or side-slip (*submodes 4,5,9,12–15*). Are these similar movement patterns really used by animals in different ways?

To answer these questions, we examined submode relationships over time. One possibility is that each of the identified submodes represents a completely autonomous behavior. If this were the case, each submode would occur independently of submodes that come before or after. In all other cases, the statistics of submodes that tend to occur before or after a given submode can be used to identify submodes that share the same statistical relationships. Such submodes will be grouped into units we will call 'modes.'

### Submode sequences demonstrate limited memory

To measure the extent of submode relationships over time, we first measured how well the identity of submodes at one time predicts the identity of their neighbors in the same trajectory. Velocity trajectories were converted to submode sequences by classifying each $v_R$ peak and the surrounding trajectory fragment, using parameters found in the previous analysis (Materials and methods). For each trajectory $j$, we obtained a submode sequence $\{u_i\}^{(j)}$, where $u$ represents submode identity, and $i$ indexes time. Among all trajectories, we found that over a quarter of the uncertainty about submode

identity $u_i$ is explained by the previous submode $u_{i-1}$, indicating that submodes are not independent ($\rho = I(u_i; u_{i-1})/H(u) = 0.271$, $CI_{95} = [0.270, 0.272]$; where $I$ is mutual information and $H$ is Shannon entropy (**Cover and Thomas, 1991**)). However, after accounting for the history of one previous submode, almost none of the remaining uncertainty was explained by submodes one step further in the past, $u_{i-2}$ ($\rho_2 = I(u_i; u_{i-2} \mid u_{i-1})/H(u_i \mid u_{i-1}) = 0.015$, $CI_{95} = [0.014, 0.015]$). This result suggested that transitions between submodes depend on the current submode, and negligibly on submode history two or more steps in the past. However, it is possible that a small contribution of prior memory can accumulate over time.

To test the cumulative impact of history on submode transitions, two Markov models that neglected all but one or two-step submode history were used to generate short synthetic submode sequences that were then compared with real sequences. These two models generated nearly indistinguishable sets of submode sequences, each comparable to random sets of real sequences (**Figure 5—figure supplement 1A**). Specifically, synthetic sequences from a first-order Markov model matched almost the same number of real sequences as were matched between two random sets of real sequences (matched fraction = $0.93 \pm 0.009$, see Materials and methods), while sequences from a second-order Markov model matched real sequences only slightly better (matched fraction = $0.96 \pm 0.008$). Given that nearly equal data fractions were matched accurately by assuming either one or two submode history, we concluded that models conditioned on one previous submode are sufficient to capture most short submode sequences. As one submode but not prior history effectively influenced transitions to the next submode, this analysis argues that memory of past behavior must extinguish between the start of one submode and the start of the next. Hence, while individual submodes represent unique temporal correlations in velocities, submodes also correspond to periods during which memory of past behavior is effectively extinguished. In this way, velocity associations over short periods captured by submode identities also defined behavior elements that are decoupled over longer periods.

## Statistical relationships between submodes suggest groups of related submodes

Next, we asked whether the statistics of submode transitions suggested additional levels of behavior organization. In principle, there are two possibilities. Statistical relationships shared by more than one submode could indicate that these submodes are interchangeable in the context of other submodes, and therefore may be grouped, or transition statistics may indicate that more than one behavior underlies a single submode. To allow for both possibilities, we used the framework of Hidden Markov Models (HMMs) to identify statistical patterns in submode sequences. Different HMMs were constructed and evaluated using multiple criteria while systematically varying the assumed number of hidden states (Materials and methods). For each number of hidden states, multiple models were trained from random initial conditions to fit observed submode sequences. All models assumed that state transitions depend only on the current state, approximating history dependence in submode transitions. Models of equivalent complexity were evaluated using the likelihood attributed by each model to real submode sequences. Models of different complexity were compared using a variant of sampling from a generative model, testing each model's ability to generate sets of individual behavior sequences that correctly represented the distribution of real trajectories (Materials and methods). By these criteria, it was found that 6-state models reproduced real submode sequences as well as models that used information about all 15 submodes, some 5-state models performed similarly to 6-state models, and all simpler models performed significantly more poorly (**Figure 5—figure supplement 1C**). Therefore, we concluded that a five state model represented a good tradeoff between model complexity and accuracy, and chose one such model based on its performance (**Figure 5—figure supplement 1E,F** and Materials and methods). This model, as all well-performing models, revealed a largely straightforward pattern: most submodes mapped predominantly to a unique underlying state (**Figure 5—figure supplement 1D**). This result was surprising as by construction, HMMs allowed each submode to be associated with multiple states. As most submodes mapped to only a single state, some submodes were associated with the same state because there were more submodes than states. Submodes that mapped to the same state could be considered a group based on their shared statistical properties. An exception to this pattern was behavior at low velocities and stops (*submode 8*, and one or more of the rare *submodes 4,5,9,14,15*), which showed signatures of potentially mixed behavioral states (see Materials and methods and below). However, where

evaluated, division of submodes into more underlying processes increased model complexity without substantially improving fit to data (see below). As many different models were evaluated, we observed that groupings of the most frequent submodes agreed between the best Markov models but submodes that occurred rarely in trajectories could be assigned to different groups, presumably reflecting their modest contributions to the dataset. As a result, a small number of models that grouped rare submodes in different ways performed comparably well by all model selection criteria, and could not be differentiated further (*Figure 5—figure supplement 1C,E*). In addition, transition probabilities below approximately 0.005 could not be accurately determined, due to finite dataset sampling error (see Materials and methods). Nevertheless, as a 5-state Markov model could generate sets of synthetic behavior sequences that matched the distribution of real sequences nearly as well as more complex models, we concluded that five submode groups, or modes, provided a sufficient model of submode organization (*Figure 5—figure supplement 1A,E*, Materials and methods).

How are submodes organized? Multiple submodes may map to the same mode because each occurred between the same set of other movements, with similar transition probabilities. Provided their statistics are well-sampled, either similar or distinct movements may be legitimately grouped by these criteria. Were similar or distinct movements grouped in modes? Each mode, like each submode, can be described as a distribution in four dimensions: the three-component velocity vector $\vec{v}$, plus time. We call this distribution a velocity profile. Qualitatively, the velocity profiles of most submodes assigned to the same mode are similar (*Figure 4*), suggesting modes represent major groups of movements.

Movement patterns represented in two of the modes were confined to either low or high ranges of $v_R$ and $v_T$ (*Modes I, V* respectively, *Figure 5A,E*). Velocity profiles of the other three modes all spanned low and high velocities in multiple velocity components (*Modes II-IV Figure 5B–D*). Movement patterns in these modes included: sharp and shallow turns accompanied by forward acceleration (*Mode II*), various side-slip motions at slow speeds, often but not always preceded by decelerations (*Mode III*), and slower, relatively straight runs with occasional side-slip (*Mode IV*) (*Figure 5B–D*, *Figure 4*). Individual movement patterns assigned to a mode could be largely judged by human observers to represent variations on a theme (*Figure 4*, *Figure 4—figure supplement 1*, and pers. obs.).

However, both unexpected groupings and distinctions were found for statistically well-represented movement patterns. Fast, straight walking was grouped with smooth turning in the high velocity *Mode V*, while intermediate and slow straight walks, with occasional side slip, were grouped together in a different mode, *IV* (*Figure 5D*). That is, in this segmentation, two apparently different behaviors, some turns and straight runs, were grouped together, and were distinguished from other types of related behaviors. On the other hand, two sharp turns (*submodes 6* and *7*) differed only subtly, but were never assigned to the same mode in any well-performing Markov model we evaluated. These groupings contrast with previous behavior segmentation efforts, where turns were deemed by human observers to be separate, a priori, from straight runs of all speeds, and sharp turns were treated as a single category (*Pierce-Shimomura et al., 1999*; *Branson et al., 2009*; *Kain et al., 2013*; *Geurten et al., 2014*). Crucially, in our segmentation distinct movements such as turns and straight walks of *Mode V* were not grouped because they tended to occur one after the other, but because they occurred interchangeably between other movements. Statistical relationships therefore also revealed associations beyond movement similarity. These relationships, the transition statistics between modes, showed an additional distinction. *Modes I* and *V* had high self-transition probabilities, and did not communicate with each other directly at significant frequencies (*Table 1A*). Rather, transitions between *Modes I* and *V* typically required passage through one or more of the intermediate *Modes II-IV* (*Figure 5H*). Due to this transition structure, *Modes I* and *V* showed longer residence times than other modes (*Figure 5G*).

A critical prediction of the Markov model used to capture mode transitions is that the dwell time distribution describing the time flies spent in each mode should be fit by a single exponential. An excellent fit between real and model-generated dwell time distributions shows that the model captures most transition probabilities in all modes, although some slow transitions in modes *I* and *III* are missed (*Figure 5G*). Because *mode III* occurred infrequently (accounting for 7.3% of data), and its dominant, fast component is well captured by the model (*Figure 5G*) it was not segmented further. Two components in the stopped mode (*I*) dwell time distribution were separated using a Hidden Markov model, capturing observed dwell times that varied over 2–3 orders of magnitude

**Table 1.** Transition rates $[Tr]_{ij}$, dataset S (uniform illumination, test tubes). Rows: 'From' mode; Columns: 'To' mode. Transitions below sampling error are gray.

|  | I | II | III | IV | V |
|---|---|---|---|---|---|
| I | 0.7764 | 0.1572 | 0.0608 | *0.0037* | *0.0020* |
| II | 0.0058 | 0.2010 | 0.0521 | 0.5598 | 0.1812 |
| III | 0.3129 | 0.3464 | 0.3082 | 0.0227 | 0.0098 |
| IV | *0.0029* | 0.0353 | 0.0349 | 0.7111 | 0.2159 |
| V | 0.0248 | 0.1298 | 0.1364 | 0.3422 | 0.3668 |

(*Figure 5G*). However, the additional stopped mode ($I_B$) that the HMM uncovered only communicated with other stops (mode $I_A$), and did not improve model performance quantitatively (data not shown). We therefore did not consider this additional mode in our subsequent analysis, and infer that this second stopping mode, $I_B$, likely represents a generally non-responsive state, such as sleep (*Shaw et al., 2000*; *Hendricks et al., 2000*). However, stopped flies also engage in several non-locomotor behaviors such as eye, wing, and abdomen grooming, oviposition, and defecation that our dataset was not designed to resolve. These behaviors would then comprise subsets of *mode I*.

The set of observed behaviors, characterized as submodes, is also described more coarsely by the smaller number of modes, potentially eliminating information. However, in sequences, uncertainty about mode identity is explained as well by the previous mode as uncertainty about submode identity is explained by the previous submode (mode $\rho = 0.303$, $CI_{95} = [0.302\ 0.304]$). Conversely, as with submodes, almost none of the remaining uncertainty is explained between modes two steps apart (mode $\rho_2 = 0.0069$, $CI_{95} = [0.0066\ 0.0072]$). By contrast, when submodes are grouped arbitrarily into five elements, information can both decrease between adjacent elements in a sequence, and increase spuriously between elements two steps apart (*Figure 5—figure supplement 2*). These results argue that grouping submodes into modes neither lumped behaviors in a way that eliminated information, nor scrambled existing relationships.

We next sought to determine the extent to which the structure of our model was influenced by the structure of our chamber, or by trajectory length. We therefore examined whether submode segmentation, groupings, and transition structure remained adequate in another dataset collected under identical illumination, but in a large arena where longer individual trajectories were obtained (Dataset L). Altogether, stereotyped patterns were broadly preserved between datasets, as the same segmentation criteria unmixed trajectory coefficient distributions from either dataset in the same independent components at all steps (*Figure 6*). At a more detailed level, we quantified velocity profile similarity using an intuitive measure of distribution overlap, the Bhattacharyya coefficient $B$ that ranges from 0 (no overlap) to 1 (complete overlap) (*Kailath, 1967*, Materials and methods). Overlap was high between the same mode types segmented in different datasets (median overlap $B = 0.976$, $P_{95} = [0.942\ 0.982]$). By comparison, overlap was lower between different modes both within and between datasets (median $B = 0.642$, $P_{95} = [0.190\ 0.822]$). Comparable results were obtained using other metrics of similarity, as well as in a third dataset where individual flies were recorded in isolation (data not shown). Thus, modes retained over 90% identity and remained distinct, with only small shifts in their velocity profiles, across different datasets.

Because trajectories are continuous and different modes must connect with each other as the fly moves, a mode's velocity profile could include parts of transition periods between modes. Velocity profiles may thus change because the distribution of adjacent modes changes. In addition, velocities may also be modulated independently of mode structure. To investigate these possibilities, we compared transition probabilities and velocity profiles during transition periods in datasets S and L, representing two conditions. Indeed, although mode connectivity remained largely the same, transition probabilities differed between conditions (*Figure 6—figure supplement 1*). As modes were defined by classifying trajectory fragments around $v_R$ peaks, in this segmentation transition periods occurred in the intervals between $v_R$ peaks. Accordingly, we compared velocity profiles of the 25 possible interval types between the five modes from datasets S and L. Of these interval types, 22 were sufficiently well sampled to compare (*Table 1*), and of these, 19/22 showed velocity profiles that were

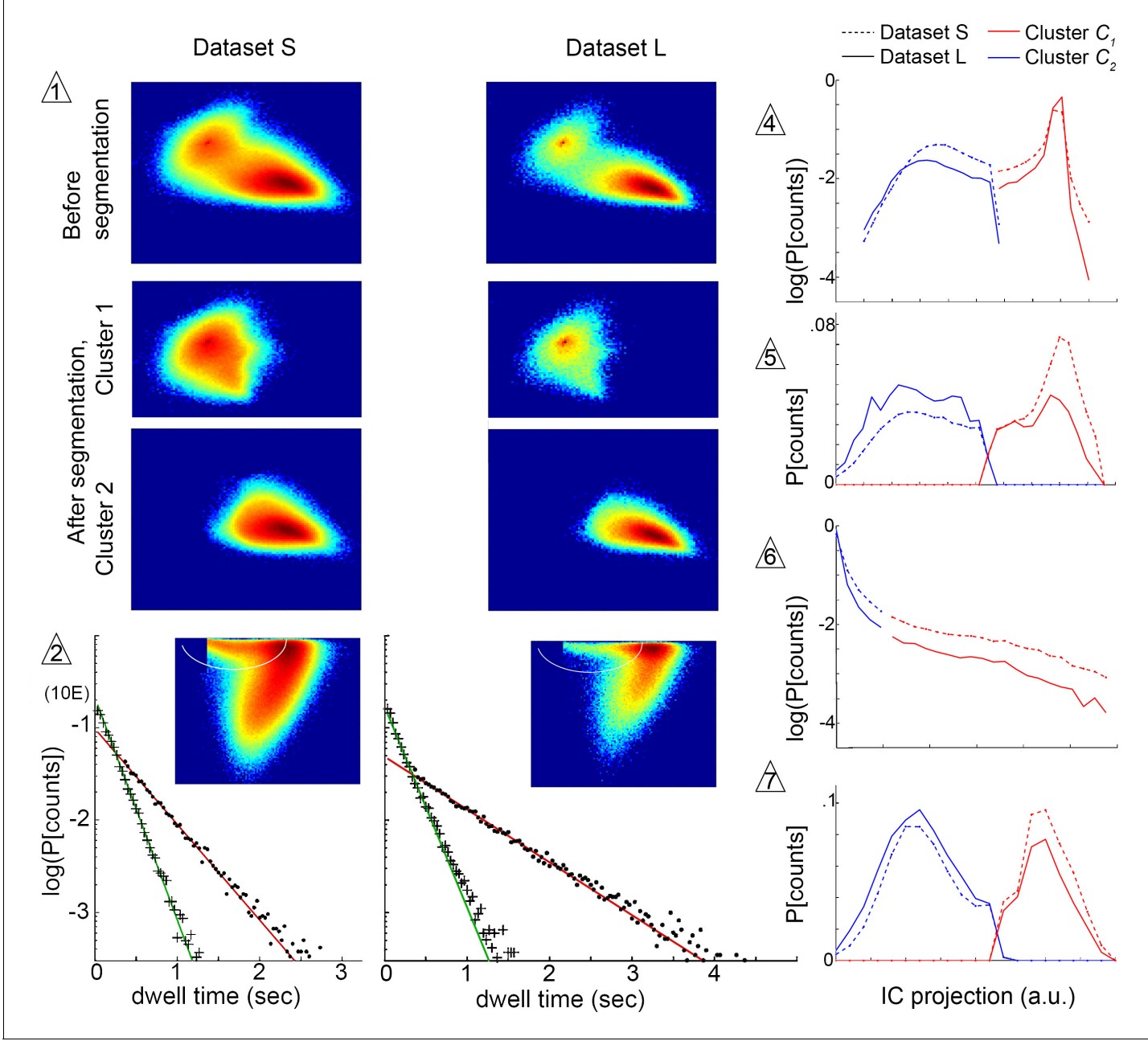

**Figure 6.** Submode segmentation in two datasets, S and L. Histograms of trajectory coefficients $\mathbf{s}^{(j)}$ in independent component space (*Box 1* and Materials and methods) are shown at decision points of the iterative segmentation procedure. Heat maps show counts in two dimensions of the independent component space, and line plots show counts in one dimension. Decision points (numbered triangles) correspond to decision points labeled on the segmentation tree in *Figure 3B*. Subsets of coefficients separated at each decision point are shown in the same dimensions, as noted. Plot dimensions do not necessarily correspond to independent components, as sometimes data was rotated in IC space to visualize multimodal coefficient distributions. Trajectories from two different experimental conditions, corresponding to datasets S and L, were projected in the same independent components, and their coefficients are shown on the same or adjacent plots, as noted below. The initial decision point (1) and all terminal decision points (2,4–7) are shown. The coefficient distribution at decision point three is not shown because it could not be represented in two dimensions. Decision pt. (1). $\mathbf{s}^{(j)}$ histograms in two dimensions, showing the multimodal distribution of coefficients from the entire dataset (Top plots), and coefficient clusters 1 and 2 after they were separated at this decision point (Bottom plots). Decision pt. (2) Data projection in two dimensions before segmentation, showing the segmentation cut (white oval), and the dwell time distributions for each resulting submode, along with single exponential fits. Note that dwell time distributions are consistent with one dominant component for each segmented submode, despite differences in the joint velocity distributions and dwell times between datasets S and L. Decision pts. (4-7) $\mathbf{s}^{(j)}$ histograms in one dimension, as used for segmentation. Note that the distribution at decision point six is plotted on a log scale, where segmentation separates the long tail. Decision pt. seven is upstream of

*Figure 6 continued on next page*

*Figure 6 continued*

further segmentations that were not validated by Markov models. As these submodes are very low frequency, they may represent distinct submodes that can be validated using larger datasets. The 5-state MM5 groups submodes as follows:

*Mode I* : $u_8$; *Mode II* : $u_7, u_3, u_{10-13}$; *Mode III* : $u_9, u_4, u_5, u_{14-15}$; *Mode IV* : $u_2$; *Mode V* : $u_1, u_6$.

The following figure supplement is available for figure 6:

**Figure supplement 1.** Model connectivity and transition frequencies in two datasets.

statistically distinguishable between conditions (**Table 2**, **Table 3**, Materials and methods). However, interval velocity profiles of the same type overlapped to a high degree, even when they differed statistically (median overlap $B = 0.96$, $P_{95} = [0.87 \ 1.0]$). Hence, considering both velocity profiles and mode connectivity, model structure was broadly similar between two different experimental conditions. At the same time, transition probabilities between modes changed with condition, and velocity profiles differed subtly regardless of how they were measured. While modes can be reliably identified under different conditions, subtle velocity profile shifts may reflect altered transitions between modes, behavioral changes independent of mode structure, or both.

How well does a Markov model that ignores velocity profile shifts and lingering correlations between submodes predict real behavior? Can real, extended behaviors be reproduced by a model that abstracts velocity trajectories as modes connected by Markovian transitions? To test the model, we measured matches between real mode sequences observed in trajectories, and synthetic sequences produced by the Markov model, while varying sequence length. Real sequences were obtained by classifying sequences of $v_R$ peaks and their surrounding trajectory fragments in trajectories from dataset L according to the 5-mode model (Materials and methods, **Figure 5—figure supplement 1**). Synthetic sequences were sampled from a generative model, the 5-state Markov chain representing mode transitions (**Figure 5H**). As in previous analyses, sets of both real and synthetic sequences can be considered a random sample of all sequences flies, or the model, might generate. Accordingly, model performance was assessed by resampling sequences from real and synthetic sequence sets, then measuring the average number of exact matches between synthetic and real samples, normalized by matches between two real samples (Materials and methods). As before, this procedure was quantified by the match fraction $f$, ranging from 0 when synthetic sequences differ completely from real ones, to one when synthetic and real sequence sets are indistinguishable. Measuring $f$ at increasing sequence lengths, we defined $\tau_P$ as the time interval over which $f \geq 0.9$, corresponding to sequence lengths over which the model performed accurately. Remarkably, the Markov model remained accurate over seconds ($\tau_P = 4.3 \pm 0.55$ s, **Figure 7A**). By comparison, velocity autocorrelations nearly vanish by 1 s (**Figure 2F**) and similar velocity patterns diverge throughout velocity space after about 200 ms (**Figures 2D** and **7B**). The Markov model thus reproduced samples of individual behavior beyond the time scale when patterns in velocity trajectories remain similar enough to be predictable. In fact, the Markov model was accurate across the entire time span of behavior that was

**Table 2.** Comparison of transition velocity profiles between datasets S and L. Overlap between mode velocity profiles is shown by transition type, comparing modes in datasets S and L. 1 = perfect overlap; 0 = no overlap. Rows: 'From' mode; Columns: 'To' mode. Gray values fail significance criteria as reported in Table 3.

|     | I    | II   | III  | IV   | V    |
|-----|------|------|------|------|------|
| I   | 1.00 | 0.97 | 0.96 | *0.83* | *0.83* |
| II  | *0.76* | 0.94 | 0.89 | 0.93 | 0.96 |
| III | 0.99 | 0.95 | 0.95 | 0.87 | *0.82* |
| IV  | *0.89* | 0.94 | 0.95 | 0.94 | 0.97 |
| V   | *0.99* | 0.96 | 0.97 | 0.97 | 0.99 |

**Table 3.** Probability that dataset L and S transitions are drawn from the same velocity profiles, by bootstrap for each transition type. $p<0.002$ is significant at *Prob*<0.05 level, using Bonferroni correction for 25 independent comparisons.

|     | I      | II     | III    | IV     | V      |
| --- | ------ | ------ | ------ | ------ | ------ |
| I   | 0.0001 | 0.0001 | 0.0001 | 0.2354 | 0.4162 |
| II  | 0.1739 | 0.0001 | 0.0001 | 0.0002 | 0.0001 |
| III | 0.0001 | 0.0002 | 0.0002 | 0.0003 | 0.0901 |
| IV  | 0.0138 | 0.0001 | 0.0001 | 0.0001 | 0.0001 |
| V   | 0.0284 | 0.0001 | 0.0001 | 0.0001 | 0.0001 |

well sampled in our trajectory datasets, raising the possibility that it may also describe walking behavior over longer periods.

Did the model capture the range of observed individual behaviors? In principle, a model might perform well simply by capturing the most common transitions in individual trajectories, such as self-transitions of stops and high-speed runs and turns (*modes I* and *V*), and matching only common behaviors like extended stop or run episodes. If this happened, matches evaluated in the generative test could be skewed by common behavioral sequences. To address this possibility, frequencies of real and model-predicted mode sequences were compared over increasing sequence lengths up to $\tau_P = 4.3$ s. Importantly, the observed frequencies of individual sequences in real trajectories were reproduced in the model-generated sequences, and thus the model did not simply describe behavior on average (*Figure 7C*). Moreover, as common mode sequences were no better matched than rarer ones, model accuracy did not depend on a few overrepresented trajectory types (*Figure 7C*). Finally, because frequencies of all real mode sequences above the sampling error in our data were matched by the Markov model with a Pearson Coefficient of R = 1.00, we conclude that the model captured all measurable transition types. Ten of these mode sequences were selected randomly and five random samples of each sequence type are shown in *Figure 7—figure supplement 1*, along with samples of the most frequent sequence type, repeats of the high-speed runs and turns *mode V*. In sum, velocity patterns were distinguished by the same segmentation criteria on different conditions and Markov models captured transition probabilities specific to each condition, sufficiently to predict individual behavior sequences at least an order of magnitude longer than the extent of locally correlated velocity patterns.

Finally, we asked whether velocity trajectories become predictable over longer periods once mode identity is known. Starting from small neighborhoods in velocity space preceding each of the five modes, trajectory divergence was measured separately for each mode. We found that after the same short period over which unsegmented trajectories remain nearby, divergence in velocity space is unaffected by segmentation (*Figures 2D* and *7B*). Whether segmented by mode identity or not, similar trajectories diverged from <0.004% of velocity space 100 ms before $v_R$ peaks, near mode initiation, across 15–20% of velocity space over the following 200 ms, during mode execution, and throughout velocity space afterwards (*Figure 7B*, *Figure 7—figure supplement 2*). As unbounded divergence occurred at the average time of behavior adjustments, mid-way between velocity peaks or modes, this finding raises the possibility that divergence is linked with apparent memory extinction captured by Markov models between the start of one mode and the start of the next. Intriguingly, bounded but significant divergence within modes also raises the possibility that a mechanism for memory loss can be intrinsic to mode structure: first order Markovity may be approximated as trajectory history is scrambled over the course of one mode (*Figure 7D,E*).

## Discussion

This work examined dynamics of walking in a relatively complex animal, the fruit fly, using principled criteria. We find that fly movements naturally decompose into different time scales, identify major movement patterns, and characterize properties of these patterns over time. On a fast time scale, distinct movement patterns emerge from patterns of variation in trajectory fragments. Each

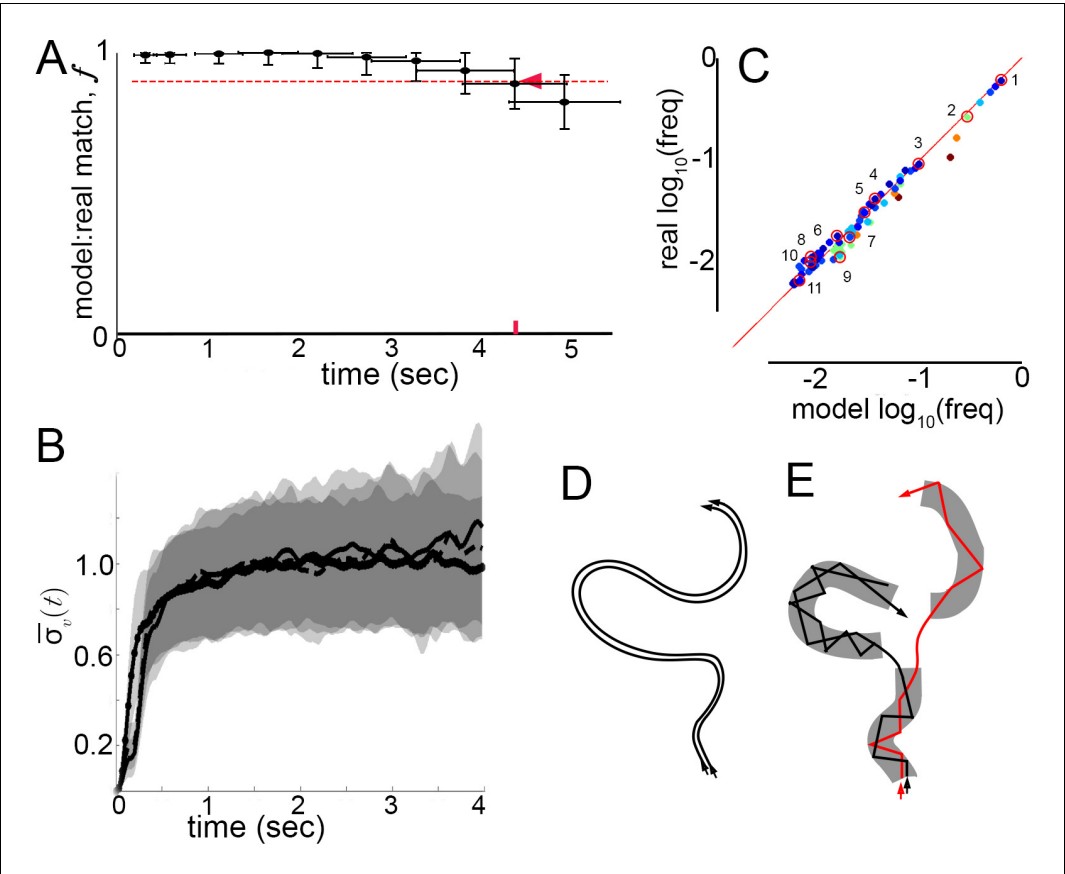

**Figure 7.** Model prediction and trajectory divergence over time. (**A**) Matched fraction ($f$) of mode sequences from real trajectories as a function of sequence length. Synthetic mode sequences were generated using the transition component of the model MM5. Red dashed line marks $f = 90\%$, red tick and arrow mark the sequence length when $f$ first crosses below 90%. (**B**) Trajectory divergence in velocity space from randomly sampled small neighborhoods, without and with information about current mode identity. Two divergence curves from *Figure 2D* are replotted over a longer time scale: starting from neighborhoods anywhere in velocity space, and including all trajectories (dot-solid line), or including only trajectories that attained a turn peak 100 ms later(dashed line), or only those with a turn peak 100 ms later corresponding to a given mode (solid line). Dashed and solid lines overlap for their durations. Note the short $\sigma$ plateau, and rise after approximately 200 ms. (**C**) Predicted versus actual log-frequency of all mode sequences above measurement noise, up to trajectories that include 17 consecutive modes, covering 4.3 s. Markers are colored by sequence length, from shortest (2-mer, dark blue) to longest (17-mer, dark red). $R = 1.00$, $n = 60$, $p \ll 0.001$; Sampling error cutoff is at $freq < \frac{1}{2}(2.58N^{-\frac{1}{2}})$, $N$ = number of sequences in each condition. Ten of these sequences (#2–11) were randomly selected for display in *Figure 7— figure supplement 1*, plus the most common sequence (#1). (**D**) Illustration of velocity trajectories specified by a stable dynamical system. Two trajectories starting from similar velocities (arrowheads, bottom), follow similar paths over time that show little divergence some time later (arrowheads, top). (**E**) Illustration of mode structure and transitions in velocity space. Trajectories from a small region of velocity phase space (black and red arrows) display bounded divergence for the duration of a mode (gray area) but display approximately no memory of past behavior at transitions (gaps between gray regions) beyond their current mode. Note divergence in the course of mode traversal may lead to path scrambling inside modes (jagged lines).

The following figure supplements are available for figure 7:

**Figure supplement 1.** Examples of common mode sequences.

**Figure supplement 2.** Trajectory divergence in velocity space, measured from the center or periphery of each mode's velocity distribution.

movement pattern corresponds to a continuum of behaviors that can be described by a common set of parameters, representing stereotyped, but not invariant behaviors. On a longer time scale, sequences of these patterns can be approximated as a stochastic, finite-memory process. In all, this description captures a dynamic structure of behavior spanning tens of milliseconds to seconds.

## Defining behavior structure from dynamics

Our behavior segmentation criteria were designed to identify subsets of trajectories that showed common dynamics within a subset, and mutually independent variation between subsets, consistent with signatures of distinct motor planning or control. Segmented by these criteria, some elementary behaviors were behaviors picked out by human observers or prior automated segmentation strategies, while others revealed surprising associations or distinctions between movement patterns. For instance, behaviors we termed submodes identified smooth turns, sharp turns, and sideways movements, all of which can be picked out by human observers (*Branson et al., 2009*; *Kain et al., 2013*), pers. obs.). Less intuitively, sharp turns that differed only subtly were distinguished as different behaviors (*Figure 4*: *submodes 3* or *7* vs. *6*), while a range of behaviors from turns to straight walks were associated in a single behavior element (*Figure 4* and *Figure 4—figure supplement 1*: submode 1). Consistently, when statistical relationships between segmented elements were examined, subtly different sharp turns remained separate (in *modes II* vs. *V*), and turns remained grouped with some straight runs but not others (*modes IV* vs. *V*, *modes II* vs. *III*). By comparison, human observers have recognized sharp ('saccadic') turns as a distinct behavior, but had not recognized the spectrum of different turn types, and had culled turns from straight runs as different behaviors a priori (*Croll, 1975*; *Pierce-Shimomura et al., 1999*; *Branson et al., 2009*; *Kain et al., 2013*). Likewise, so had behavior segmentation strategies that largely removed human judgment but parsed behavior in instantaneous observable parameters, by thresholding fast versus slow velocities, or by segmenting orthogonal (mutually exclusive) animal postures (*Braun et al., 2010*; *Geurten et al., 2010*, *Geurten et al., 2014*). These simple examples underscore how differently behavior dynamics may be organized relative to behavior features salient to human observers, or machine classification that collapses time scales.

Contrary to these segmentation strategies, our criteria identified distinct behavior patterns that overlap substantially in velocity space (*Figure 5F*). Because they overlap, these patterns cannot be separated from each other solely due to biomechanical constraints. At the same time, these patterns are unlikely to represent mere extremes of a behavior continuum (*Gallagher et al., 2013*; *Szigeti et al., 2015*), because they show mutually independent variation and approximately Markovian statistical relationships over time.

We hypothesize that dynamically independent behaviors in our segmentation reflect differences in neural control. While no comparable data yet exist in walking *Drosophila*, dynamics of global neural activity in *C. elegans* suggested that worm locomotion includes multiple, distinct turn types and forward run states that had not been uniquely identified in previous analyses of worm behavior (*Kato et al., 2015*). In worms, however, behaviors have not yet been defined directly from the dynamics of locomotion. It remains to be seen whether neural and behavioral states correspond when both are defined from dynamics.

## Measurement

Several approaches to behavior segmentation have been developed to date, using different measurements that include parameterized models of limb or whole-animal posture (*Fod et al., 2002*; *Stephens et al., 2008*), two- or three-dimensional images of postures (*Berman et al., 2014*; *Wiltschko et al., 2015*), measures of gait (*Kain et al., 2013*), and whole-body velocity (*Braun et al., 2010*; *Geurten et al., 2010*). The most comprehensive measurements could seem like the best starting point for unbiased segmentation, but which measures of behavior are informative depends on behavior organization, itself unknown. Since similar goals can be reached using different movements, and similar movements can be produced by different muscle actions (*Bernstein, 1967*; *Winter, 1984*), an increase in measured parameters does not guarantee a proportional increase in behavior resolution. On the other hand, to find behaviors from first principles requires searching potential combinations of parameters and their dynamics. This search space expands with the number of parameters. To aid search, two strategies have been applied. In one, the space of

instantaneous postures is decomposed first, and then, in a reduced space of posture elements, dynamics are fitted to indirect measures of behavior (*Stephens et al., 2008*, *2010*). In other approaches, indirect measures of dynamics are included with posture measurements (*Berman et al., 2014*; *Wiltschko et al., 2015*). In both cases, the representation of either behavior or its dynamics is indirect at segmentation.

For this work, our aim was to analyze behavior directly from dynamics. For this reason we chose behaviors that could be represented by a small number of parameters. Walking Drosophila show multiple gaits (*Strauss and Heisenberg, 1990*), but gait use is correlated with velocity, gaits transition smoothly with velocity changes (*Mendes et al., 2013*), and movements of different body segments are tightly correlated during walking maneuvers (*Geurten et al., 2014*; *Fujiwara et al., 2017*). For these reasons, and since body displacement is the goal of locomotion, measurement of body velocities over time can provide sufficient resolution for locomotor behaviors in flies. However, many animals use more complicated limb or body segment movements during locomotion, and non-locomotor behaviors require measurements of posture (*Stephens et al., 2008*; *Wiltschko et al., 2015*). In spite of a long history of investigations, it remains to be seen in what ways movement is structured with respect to specific body configurations, action goals irrespective of configuration, or both.

## Experimental biases

Animal behavior bridges many types of external events and internal processes. Unsurprisingly, behaviors operate on multiple time scales, and many behavior properties are sensitive to environmental conditions. Fly locomotion shows trends on the scale of minutes to hours that are sensitive to age, sex, strain, satiety and other internal and environmental conditions (*Martin et al., 1999a*); pers. obs.). For instance, after removal from one type of laboratory food, locomotion decreases over tens of minutes under low illumination in both genders of different strains, but activity is gender-dependent at shorter time scales, strain-dependent at longer time scales, and illumination-dependent on the scale of 1–2 hr (*Martin et al., 1999a*). Under some conditions, fly locomotion has also been suggested to be scale-free (*Cole, 1995*). In all, fly locomotion can show trends on multiple time scales that depend on experimental conditions.

For this work, the goal was to describe properties of behavior elements under stable conditions. Therefore, experimental conditions were chosen so as to minimize trends in locomotion over the course of experiments. In these conditions, flies kept moving over 30 min and velocities approximated steady state for the entire period. Data was then collected for 10 min to stay well within this period. In addition, a few different conditions were explored but all shared the same criterion, that flies remain active without gross trends in locomotion during the experiment (see Results). By this criterion, we tried to isolate behavior properties from other potential variables controlling behavior.

Under different conditions, trends in fly behavior were found on long time scales by *Berman et al. (2014)*, *(2016)*, who attributed these trends to an internal state change. In those conditions, locomotion largely stopped shortly after removal from food and behavior was then analyzed over the following hour (*Berman et al., 2014*). As a result, over 85% of analyzed behaviors were non-locomotor movements while flies were standing, and here, Markovity was found to break down over one to a few seconds. This time scale is well-sampled in our data, where, by contrast, a generative Markov model produced excellent fits to data. In all, datasets and analyses differ between the two studies in several ways (Materials and methods). Since the majority of behaviors analyzed by Berman et al. are non-locomotor, and our studies used different genders, it is possible that non-locomotor and locomotor behaviors have different properties, or that memory in behavior is gender-specific. A likely possibility is that behavior transitions can be adjusted over time by internal or external variables (*Berman et al., 2016*), suggesting one level at which behavior structure may be modified. Notably, the time scale of elementary behaviors may be invariant to conditions, gender, or other differences between studies to date. This time scale is seen to be a few hundred milliseconds by three different methods in different datasets: increased densities in posture time series power spectra (*Berman et al., 2014*), mean residence times in inferred postural states (*Berman et al., 2016*), and trajectory divergence properties in velocity phase space (this work).

## Mode structure and finite memory in behavior

We find that temporal structure, defined by correlations in velocities over short times, implies a decoupling over longer time scales. This decoupling between modes can be described as approximately Markovian, in that transitions between modes depend on the current mode with negligible contribution from prior history. Within any one mode, similar velocities diverge, up to a bound of approximately 20% of velocity space. Around the time of transition between modes, effectively unbounded velocity divergence is observed. These findings raise three possibilities: history of prior behavior can be forgotten in the course of executing a mode, during transitions between modes, or both. While it is possible that there might be specific neural circuits devoted to maintaining and forgetting prior states (*Martin et al., 1998*, *1999b*), we note that memory loss may also be intrinsic to the mode structure we describe, or the dynamics of neural activity underlying mode execution.

## Mode structure and stability

Behavior is relatively similar any time an animal executes the same mode. Whatever the animal does beforehand, the range of potentially different behaviors that can precede a given mode must narrow to the domain of that mode. At this level, modes can be said to show a transient stability. However, our data provides no evidence that modes show asymptotic stability, under which behaviors would tend towards some typical, canonical behavior within each mode, or more abstractly, a behavior attractor. In fact, on average, originally similar velocities diverge severely in the course of mode execution, and divergence of the more typical velocity trajectories within a mode differs little from that of the less typical (*Figure 7—figure supplement 2*). If behavior can be described by attractor dynamics with noise, the attractors cannot be strong relative to noise. In all cases, similar velocities at the start of a mode are unlikely to remain similar at the time of transition to the next mode. Hence, at the level of modes, dynamic stability is a weak constraint in spontaneous behavior.

## Interface between behaviors

Many behaviors are thought to comprise sequences of relatively stereotyped movements that unfold over seconds or longer (e.g. *Marler and Hamilton, 1966*; *Baerends, 1971*; *Spieth, 1974*; *Liu and Sternberg, 1995*; *Seeds et al., 2014*). To specify an entire sequence reproducibly would require a complex, relatively high-dimensional dynamical system. This dynamical system would need to cope with internal, neural variation and potentially different initial conditions each time a behavior is to be executed. For instance, variation in neural signals, mismatched mechanical impedances, or external forces, as well as changing behavioral goals, muscle tone, or afferent feedback all change the context in which any given movement is planned and executed (*Bernstein, 1967*). These differences may persist or become amplified in the course of a sufficiently complex dynamical process unless it is carefully configured for stability. Yet, in theory, stable high-dimensional dynamical systems are rare (*Smale, 1966*).

As one solution, behaviors may be stabilized using feedback between actual and predicted movement outcomes (eg. *Todorov and Jordan, 2002*). However, some stereotyped behaviors proceed to completion irrespective of task-specific feedback (*Lorenz and Tinbergen, 1938*; *Willows, 1967*; *Barlow, 1968*). Moreover, even when movement execution can be stabilized by feedback, the template of an intended movement still corresponds to a dynamical system, whose stability remains relevant.

As another solution, complex movements may be represented in terms of simpler, independently specified components, sometimes termed primitives (reviewed in: *Flash and Hochner, 2005*), potentially composed into actions via hierarchies of motor control (*Sherrington, 1906*; *Tinbergen, 1950*). However, modular organization on its own does not guarantee dynamic stability over time. Behavior primitives must interface with each other. Likewise, so must different levels of control. Yet, when different dynamical systems interact, the history of one system can propagate to another. In this way, unintended deviations can also propagate over time. Long-term system stability is not guaranteed.

The properties of behavior elements described in this work suggest how potential challenges in building and maintaining complex, reproducible behaviors may be managed in nature. Broad velocity distributions associated with each behavior mode may accommodate a range of initial conditions, allowing different kinds of behaviors to follow one another. At the interface between behaviors, sensitivity to initial conditions can be minimized by memory loss observed over the course of single

behaviors. In this way, subsequent behavior transitions can be insulated from past ones. Over longer times, finite memory between behavior elements can bound susceptibility of behavior to instability, allowing construction of more complex behavioral sequences. Interestingly, human and other vertebrate movements decompose into short episodes with a single velocity extremum following stroke or ablations, fusing gradually into smooth movements in the course of recovery; this suggests that normally smooth movement in vertebrates may be composed from such sub-movements (*Rohrer et al., 2002*, *2004*; *Stein, 2008*). To what extent behavior properties found in this work are also found in other contexts and organisms remains to be investigated. This work suggests one possibility for how animals maintain dynamic complexity but limited susceptibility to consequences of any one behavior.

## Materials and methods

### Datasets

A lab strain of wild-type Oregon-R *D. melanogaster* with homology to Oregon-R-mod-ENCODE (RRID:BDSC_25211; D. Gohl, pers. comm.) was used for all data collection. Single fly trajectories were obtained as described in *Katsov and Clandinin (2008)* and are available at the Dryad Digital Repository (*Katsov et al., 2017*). Briefly, the centroid location and orientation of individual flies walking alone or in groups under uniform illumination from below (5.5 cd/m2) in an otherwise dark room were recorded at 30 frames per second. Flies walked either in long test tubes (25 × 150 mm, VWR 89001–458; Datasets S, A) or a large, custom-built arena (300 mm diameter; Dataset L). Behavior was recorded for 10 min under conditions where locomotion remained near steady state up to 30 min (*Katsov and Clandinin, 2008*). Trajectories were retained only from the top, central portion of each chamber (8 × 100 mm of test tube, 200 × 200 mm of arena). The position of the head was inferred statistically with >99.8% accuracy. Trajectories were filtered with a Gaussian kernel $FWHM \approx 0.08$ s (s.d.=1 camera frame), as in previous work (*Katsov and Clandinin, 2008*).

### Velocity phase space, construction and measurements

Construction. Approx. $10^6$ trajectories from 6930 female flies (median length 0.93 s, $P_{2.5}$-$P_{97.5}$ = [0.30, 4.5] s) were used to measure instantaneous velocity $\vec{v} = (v_T, v_R, v_S)$ and acceleration $\dot{\vec{v}} = (\dot{v}_T, \dot{v}_R, \dot{v}_S)$. These were used to construct distributions of accelerations at $N = 41615$ velocity component locations, $\Pr\left[\dot{\vec{v}} \mid \vec{v}\right]$ spanning $|v_R| < 450°s^{-1}$ in 25°s$^{-1}$ increments, $-0.6 \leq v_T \leq 3.2$ cm s$^{-1}$ in 0.09 cm s$^{-1}$ increments, and $|v_S| < 0.94$ cm s$^{-1}$ in 0.063 cm s$^{-1}$ increments. Although flies are capable of attaining velocities outside this range, these were not well sampled in our dataset and the velocity range was truncated accordingly to retain useful estimates of $\Pr\left[\dot{\vec{v}} \mid \vec{v}\right]$. The average number of acceleration data points per velocity bin was $\langle n_v \rangle \approx 500$, but taking into account autocorrelation time, the effective number of data points per velocity bin was $\langle n_v \rangle_{eff} \approx 100$, with cutoff set at $n_{eff} < 10$. Acceleration distributions spanned, $|\dot{v}_R| < 4640$ °s$^{-2}$, $|\dot{v}_T| < 16.4$ cm s$^{-1}$, $|\dot{v}_S| < 10.1$ cm s$^{-1}$. In all, 99.7% of recorded behavior was retained. The distribution $\Pr\left[\dot{\vec{v}} \mid \vec{v}\right]$ was averaged over $\dot{\vec{v}}$ at each $\vec{v}$ to produce $\mathbf{E}\left[\dot{\vec{v}} \mid \vec{v}\right]$. This expectation is a vector field, describing average changes in velocity at each combination of velocity components. Local dispersion was estimated by the mean absolute deviation, a conservative estimate of dispersion, taking expected value $\mathbf{E}\left[|\dot{\vec{v}} - \langle\dot{\vec{v}}\rangle|\right]$ over $\dot{\vec{v}}$ at each $\vec{v}$, where $\langle\dot{\vec{v}}\rangle = \mathbf{E}\left[\dot{\vec{v}} \mid \vec{v}\right]$.

Divergence of trajectories $\vec{v}(\tau)$ was measured from similar initial conditions, defined as small neighborhoods in velocity space each covering 0.004% of this space. Velocity neighborhoods were randomly chosen by sampling velocity bin indices $b \in \{1, \ldots, N\}$ corresponding to bin locations in the 3-component velocity space. A random sample of $N_B = 200$ bins was drawn. Each sampled bin $b_i$ produced a set of trajectories $\{\vec{v}(t)\}^{(b_i)}, i \in \{1, \ldots, N_B\}$, that passed through this bin at time $t_b$, with $t = \tau - t_b$. Trajectory spread at time $t$ was quantified by standard deviations $\sigma_t\left[\{\vec{v}(t)\}^{(b_i)}\right]$, averaged

over all randomly sampled neighborhoods and normalized to the standard deviations of the time-independent velocity component distributions:

$$\sigma(t) = \frac{1}{\boldsymbol{\sigma}[v_T]\boldsymbol{\sigma}[v_R]\boldsymbol{\sigma}[v_S]} \langle \boldsymbol{\sigma}_t[v_T]\boldsymbol{\sigma}_t[v_R]\boldsymbol{\sigma}_t[v_S] \rangle_{N_B} \tag{1}$$

In the space of the 3 velocity components, $\sigma(t)$ describes the dispersion of trajectories as a fraction of the range of attainable velocities. Trajectories were sampled in four ways for different analyses. First, all trajectories were included that passed the same neighborhood at $t = 0$. Second, trajectories were selected that passed the same neighborhood at $t = 0$ then attained a $v_R$ extremum at different times afterward, $t = [67, 100, \ldots, 300]$ ms. The delay $t = 100$ ms is half of the average interval between $v_R$ extrema. Third, trajectories were selected that passed the same neighborhood at $t = 0$, attained a $v_R$ extremum at $t = 100$ ms, and were classified as a specific mode $m$ at that time. Fourth, trajectories were selected that passed the same neighborhood at $t = 0$, attained a $v_R$ extremum at $t = 100$ ms, were classified as mode $m$, and fell in the same quartile of this mode's velocity profile at $t = 0$, $\Pr[v_T, v_R, v_S]^{(m)}$, $m \in \{I..V\}$. 95% confidence intervals were obtained by repeating the procedure 50 times to estimate the error of the mean.

## Submode segmentation

### ICA setup

As described in **Box 1**, times $\{t_e\}$ of local extrema in $v_R(\tau)$ were identified over $\tau \pm 83.3$ ms ($\pm 2$ camera frames), and trajectory fragments $\vec{\boldsymbol{v}}(t'), t' = \tau - t_e$, covering $n$ camera frames were isolated around each $v_R(t_e)$; initially $n = 33, |t'| < 550$ ms, but see below. Fragments $\vec{v}(t')$ were represented as $(3n \times 1)$ column vectors concatenating the 3 velocity components, normalizing turn direction and preserving relative side-slip direction:

$$\mathbf{v} = \begin{bmatrix} cv_R(t') \\ v_T(t') \\ cv_S(t') \end{bmatrix}, \quad c = sgn[v_R(t' = 0)]. \tag{2}$$

Then, these vectors $\mathbf{v}^{(j)}, j = [1, \ldots, N]$, were concatenated horizontally and standardized row-wise:

$$\mathbf{M} = \begin{bmatrix} \mathbf{v}^{(1)} & \mathbf{v}^{(2)} & \cdots & \mathbf{v}^{(N)} \end{bmatrix} \tag{3}$$

$$\mathbf{Z} = \begin{bmatrix} \mathbf{z}^{(1)} & \mathbf{z}^{(2)} & \cdots & \mathbf{z}^{(N)} \end{bmatrix}, \quad z_i^{(j)} = \frac{v_i^{(j)} - \langle v_i^{(j)} \rangle_j}{\sigma[v_i]} \tag{4}$$

We followed *Hyvärinen and Oja (2000)*, *Hyvärinen (1999)* and *Himberg et al. (2004)* to identify robust convergence of the FastICA algorithm for decomposition $\mathbf{Z} = \mathbf{AS}$, where $\mathbf{A}$ is a $(3n \times D)$ mixing matrix and $\mathbf{S}$ is a $(D \times N)$ matrix that contains $D$ independent component coefficients of sample $j$, $\mathbf{s}^{(j)}$, in each column. This procedure was done on the entire dataset in the first iteration, $N \approx 2.9 \cdot 10^6$ trajectory fragments, and then iteratively on subsets as described below.

Treating trajectory fragments $\mathbf{z}$ as mixtures of components $\mathbf{s}$ in independent components basis $\beta$, the generative model is reformulated using demixing matrix $\boldsymbol{W}_\beta$ as: $\boldsymbol{S} = \boldsymbol{W}_\beta \boldsymbol{Z}$. Taking a row $\mathbf{w}_i$ of the demixing matrix, $\mathbf{s} = \mathbf{w_i z}$ represents a coefficient of fragment $\mathbf{z}$ in independent component $i$. Independent components are found using the expectation that coefficient distributions tend to become Gaussian when $\mathbf{wz}$ projections represent sums of independent variables, and become less Gaussian as independent variables are demixed. FastICA searches for $\mathbf{wz}$ projections distributed the least like Gaussians. The FastICA algorithm was used to maximize $\mathbf{E}[G(\mathbf{wz})]$, where $G$ is a contrast function. We used $G \approx kurtosis$, in part to minimize computation time (*Hyvärinen, 1999*; *Hyvärinen and Oja, 2000*). To ensure robust convergence, the search was repeated from random initial guesses of $\mathbf{W}$ (*Himberg et al., 2004*).

### Dimensionality reduction

In practice, the search is aided by lower data dimensionality and a $\mathbf{W}$ search space limited to orthogonal transformations. The matrix $\mathbf{Z}$ (*Equation 4*) was whitened to produce

$$\mathbf{Z}^{\mathbf{w}} = (\mathbf{E}\mathbf{D}^{\frac{1}{2}}\mathbf{E}^{\top})\mathbf{Z}, \tag{5}$$

where $\mathbf{D}$ is a diagonal matrix of the eigenvalues of $Cov[\mathbf{z}, \mathbf{z}'] = \frac{1}{N}\mathbf{Z}\mathbf{Z}^{\top}$, and $\mathbf{E}$ is the corresponding matrix of column eigenvectors.

In preliminary data analysis, it was found that principal components corresponding to time points $|t'| > 300$ ms accounted for a negligible fraction of total variance in the dataset. Moreover, when included, these dimensions slowed FastICA convergence and increased the spread of converged values of $\mathbf{W}$ without substantially affecting their mean. We inferred that these time points contributed more noise than signal and retained only $|t'| \leq 300$ ms in further analysis, corresponding to $n = 19$ camera frames per fragment.

While performing ICA on the entire dataset, it was found that FastICA convergence suffered when all 3 velocity components were included. Because $v_S$ often correlated with $v_R$, FastICA convergence was tested using fragments without any $v_S$ dimensions. This improved convergence. However, we noted that when they did converge, IC solutions found from all 3 velocity components separated data features better than any solutions from fragments lacking $v_S$. To improve ICA convergence without eliminating information contributed by $v_S$, we substituted one velocity variable in fragment matrix $\mathbf{M}$ for $v_R$ and $v_S$ : $v_H = v_R - \frac{\partial}{\partial t}(\arctan \frac{v_S}{v_T})$. This is a combined rotational velocity corresponding to velocity of heading direction, rather than of body orientation only.

Operating on the entire dataset, the first iteration of ICA was set up using modified *Equations 2– 5*. Reduced $(38 \times 1)$ vectors,

$$\mathbf{v} = \begin{bmatrix} cv_H(t') \\ v_T(t') \end{bmatrix}, \quad c = sgn[v_R(t'=0)], \quad |t'| \leq 300 \text{ ms}, \tag{2A}$$

were concatenated into (38 x N) matrices

$$\underline{\mathbf{M}} = \begin{bmatrix} \underline{\mathbf{v}}^{(1)} & \underline{\mathbf{v}}^{(2)} & \cdots & \underline{\mathbf{v}}^{(N)} \end{bmatrix} \tag{3A}$$

and standardized to produce matrix

$$\mathcal{Z} = \begin{bmatrix} \underline{z}^{(1)} & \underline{z}^{(2)} & \cdots & \underline{z}^{(N)} \end{bmatrix}, \quad z_i^{(j)} = \frac{\mathbf{v}_i^{(j)} - \langle \mathbf{v}_i^{(j)} \rangle_j}{\sigma[\underline{v}_i]} \tag{4A}$$

The matrix $\mathcal{Z}$ was then whitened to produce

$$\mathcal{Z}^{\mathbf{w}} = (\mathbf{E}\mathbf{D}^{\frac{1}{2}}\mathbf{E}^{\top})\mathcal{Z}, \tag{5A}$$

where $\mathbf{D}$ and $\mathbf{E}$ are eigenvalue and eigenvector matrices of $Cov[\bar{z}, \bar{z}'] = \frac{1}{N}\mathcal{Z}\mathcal{Z}^{\top}$, respectively.

Evaluating the entire fragment dataset, it was found that six principal components account for 87% of total variance, and only the $(6 \times N)$ projection $\widetilde{\mathcal{Z}}^{\mathbf{w}}$ into these PCs was used in the first iteration of ICA. Eliminated dimensions may contain biological noise or non-dominant contributors to total behavior variation. Non-dominant contributors were pursued by applying the procedure iteratively on data subsets after removing major components (see below).

## Iterative ICA, first iteration

When ICA converged on a solution $\mathbf{S} = \mathbf{W}\widetilde{\mathcal{Z}}^{\mathbf{w}}$ , we examined joint histograms of independent component coefficients obtained in $\mathbf{S}$, $\{\Pr[\mathbf{S}_{\mathbf{i}_1}, \mathbf{S}_{\mathbf{i}_2}]\}, i_n$ specifying a row index, $i_1 \neq i_2$. Each row in $\mathbf{S}$ corresponded to coefficients in one independent component for all trajectory fragments, and rows were examined pairwise for simplicity. After one ICA iteration, at least three clusters in the 6-dimensional space of $\mathbf{S}$ were apparent by visual inspection. As distributions with clear clusters are non-Gaussian at least in some dimensions, these components may have maximized the contrast function $G$ without necessarily being independent. For this reason, clearly non-Gaussian features of IC projections were segmented iteratively. Statistical dependence of segmented components was subsequently assessed explicitly (below).

To begin the iterative procedure, the biggest cluster in $\Pr[\mathbf{S}]$ was separated from smaller ones along the minimum data density between these clusters (*Figure 3B*, decision point 1). This

segmentation turned out to approximately divide trajectories into patterns that graze near-zero $v_T$ (Cluster 1) and those that do not (Cluster 2).

## Segmentation iteration

The ICA procedure was repeated on each cluster. Matrix $\mathbf{M}$ (*Equation 3*) was separated into $\mathbf{M}^{(C_1)} \text{ and } \mathbf{M}^{(C_2)}$, $C_1$ and $C_2$ corresponding to column indices of velocity fragments belonging to Cluster 1 and Cluster 2, respectively, when projected to $\mathbf{S}$. The rest of the procedure was repeated on each $\mathbf{M}$ subset as described above, including row-wise standardization, PCA, and ICA repeated from random initial guesses of demixing matrices $\mathbf{W}$. Pairwise k-means clustering was used to separate $\mathbf{S}$ features whenever appropriate (*Duda et al., 2000*). Like the first iteration, each subsequent iteration began with full 1.1 s fragments $\mathbf{v}^{(j)}$. It was found empirically, as in the first iteration, that fragment interval could be reduced in each subsequent iteration too, in some cases down to $|t'|$<200 ms. All velocity components $(v_R, v_T, v_S)$ contributed useful information for segmentation at decision points 4,7 and subsequent iterations (*Figure 3B*). At each iteration, 5–6 PCs were retained prior to ICA accounting for >90% variance within each subset. The total variance accounted for after all ICA iterations is a weighted sum, rather than product, of variance retained by PCs at each iteration (weighted by subset size), because each round began with the raw data subsets of velocity fragments (M columns, v). Hence, the total variance accounted for at the end of the procedure is likely >87%, the lowest variance accounted for by PCs retained at any one step. Iteration was halted when further rounds failed to find ICs substantially different from those of the starting subset, when projection into ICs did not reveal clearly separable data features, or when ICA failed to converge from random initial conditions.

Cluster 1 was segmented over four further iterations into 13 submodes (*Figure 3B*, decision points 3–7). These 13 submodes comprised different movement types traversing zero or near-zero $v_T$.

Cluster 2 was segmented into two submodes, but not using ICA because further ICA on this subset did not converge on dimensions substantially different from the starting ones in $\mathbf{v}$. Nevertheless, this subset appeared to contain a mixture of movement patterns as its joint distribution $\Pr[v_T(t_e), v_R(t_e)]$ showed two tails and kinetics of exit from this subset showed at least two distinct components (*Figure 6*, decision point 2 and data not shown). Furthermore, it was found that behavior switched out of this subset by two distinct sets of velocity trajectories (data not shown). Based on these observations, Cluster 2 was segmented to separate its distinct joint velocity distribution tails and transition paths, producing submodes 1 and 2 (*Figures 3* and *6*).

## Evaluating history dependence between submodes

Individual trajectories $\mathbf{v}(t)$ from Dataset S were segmented using demixing matrices $\mathbf{W}^{(C)}$ from the iterative classification, producing submode sequences $(u)_k, u \in \{1..15\}, k > 2$, and transition probability distributions $\Pr[u]$, $\Pr[u_i \mid u_{i-1}]$, $\Pr[u_i \mid u_{i-1}, u_{i-2}]$ were measured assuming history dependence on 0 to 2 previous submodes. We then constructed artificial submode sequences drawing iteratively from these distributions and compared each set with a set of real sequences. The 'matched fraction' of real sequences, quantifying set overlap, was calculated as described below ('sampling from a generative model', and 'match test'). This fraction is the average number of exact matches between synthetic and real sets, normalized by the average number of matches between two real sets (*Equation 10*).

## Markov models

We constructed HMMs treating submode sequences as emissions. Models were trained from random initial conditions, assuming a small number of hidden states underlying the 15 previously identified submodes. Markov model assumptions were tested via three different approaches (described below), and adequate model convergence was verified when training from different, random initial conditions converged to nearly identical models. Repeatedly converged models were evaluated in two ways. Models of the same order (same number of parameters) were compared using the log-likelihood of observed data given a particular model. In addition, models of the same and different order were compared using a variant of sampling from a generative model (described below). In well-performing HMMs, almost every submode was predominantly emitted from a single major

state, with all secondary states together contributing a small fraction of total emissions per submode (*Figure 5—figure supplement 1D*). Based on these observations of HMM structure, we constructed Markov Models (MM) in which underlying states were no longer hidden, significantly reducing the number of model parameters. Moreover, MMs permitted direct estimation of model parameters from submode frequencies observed in movement trajectories, rather than by inference as in HMMs. MMs were constructed by grouping submodes into the same emission class based on their proximity in the hierarchical clustering tree (*Figure 3*), or by lumping submodes constituting a dominant emission from the same hidden state in the best performing HMMs (*Figure 5—figure supplement 1E,F*). Multiple MM variants were constructed and evaluated, comprising different MMs with 4–6 states.

## The markov model

Starting from a set of possible states and emissions, classified at $v_R$ extrema occurring at times $\{t_e\}$, where $state(t_e) \in \{q_1, \ldots, q_N\}$, $emission(t_e) \in \{u_1, \ldots, u_M\}$, we trained HMMs with N = 2 to 6 states and MMs with N = 4 to 6 states, and used all $M = 15$ submodes identified in segmentation as emissions in both cases. In the case of MMs, states $q$ correspond to submode groups termed modes, $m \in \{m_1, \ldots, m_N\}$.

## Number of model parameters

The transition probability matrix $\mathbf{T_{ij}} \in \mathbb{R}^{N \times N}$ for both HMM and MM models is given by: $[Tr]_{ij} = \Pr[state(t_{e+1}) = q_j \mid state(t_e) = q_i]$

Since the sum of each row in the matrix (corresponding to a specific pre-transition state) is equal to 1, there are $N(N-1)$ free parameters defining this matrix. The emission probability matrix $\mathbf{E} \in \mathbb{R}^{N \times M}$ is given by:

$$[E]_{ij} = \Pr[emission(t_e) = u_j \mid state(t_e) = q_i]$$

In the HMM case, the sum of each row in the emission matrix (corresponding to a specific state) is equal to 1, and hence there are $N(M-1)$ free parameters defined by the matrix. In the MM case, an additional condition is that each column of the emission matrix (corresponding to a particular submode) contains only 1 non-zero. Accordingly, $(M-N)$ free parameters define the matrix (provided $M \geq N$). Hence, while number of states and their definitions contribute the same number of parameters in HMMs and MMs of comparable size, discrete emissions contribute $O(MN)$ free parameters in the HMM case and $O(M)$ free parameters for MMs.

## HMM training

For a given model order, models were trained using 15 sets of randomly chosen initialization parameters and training data subsets. Random initial transition and emission probability matrices were generated by sampling from a uniform distribution on [0,1] and normalizing matrix rows, and 1500 short submode sequences were randomly chosen for training out of a set of 371775 sequences (mean sequence length = 3.32 emissions; s.d. 3.27). A constrained optimization procedure (using the MATLAB optimization toolbox function fmincon, with an interior-point algorithm) was used to find locally optimal parameters which maximize the log-likelihood of the training set (computed using the MATLAB statistics toolbox function hmmdecode) with the constraints being that all rows in the transition and emission matrices sum to 1.

## MM training

A set of models was assessed for each model order of 4, 5 or 6 states. Submodes were grouped based on (1) results of HMM training, (2) relationship between submodes in the classification tree, and (3) similarities between submode velocity profiles and connectivity with other submodes. The initial submode assignment to states was as follows:

 for 4 states - {8,[3 10:13],[4 5 14],1};
 for 5 states - {8,[3 10:13],[4 5 14],1,2};
 for 6 states, 3 different initial state assignments were evaluated:
 {8,[10:13],[4 5 14],1,2,3}, {8,[3 10:13],[4 5],14,1,2}, {8,[3 10:13],[4 5 14],15,1,2}.

Models were generated for all possible combinations of submode-state assignments for the unassigned submodes in each case. Maximum likelihood estimates for single and joint submode probabilities were computed from submode occurrence frequencies in the dataset, and transition and emission matrices were derived from these probability distributions:

$$[Tr]_{ij} = \sum_{u_k \in q_j; u_l \in q_i} \Pr[u_k(t_{e+1}) \mid u_l(t_e)] \tag{6}$$

$$[E]_{ij} = \frac{\Pr[u_j(t_e)]}{\sum_{u_k \in q_i} \Pr[u_k(t_e)]} \tag{7}$$

## Model comparison criteria

We computed log-likelihoods of 371775 submode sequences in the training set from Dataset S, based on each model. Ten models with the highest log-likelihood scores for each model type (4–6 states) were kept for further validation and performance testing.

### likelihood test

For each model, ten test sets consisting of 20000 submode sequences were chosen randomly from Dataset S, likelihoods for each test set were estimated given the model, and the average likelihood of all ten test sets reported as the mean likelihood.

### sampling from a generative model

First, 11 sets of 5000 submode sequences of length 5 were randomly selected from the 83746 submode sequences in the training dataset that consisted of no less than 5 submodes. This was done by first randomly selecting a sequence from all sequences consisting of more than 5 submodes, and then randomly selecting a starting index that is at least 5 submodes away from the end of the sequence. No sequence was used more than once in any of the 11 test datasets. The number of exact matches between one set and each of the remaining 10 sets was counted as described in 'Match Test' below (*Equation 8*). Then, 10 sets of 5000 sequences of length 5 were generated from each of the generative models evaluated.

For independent, and first and second order Markov models (models $I, M1, M2$), synthetic sequences were generated by iteratively sampling from submode transition probability distributions $\Pr[u]$, $\Pr[u_i \mid u_{i-1}]$, $\Pr[u_i \mid u_{i-1}, u_{i-2}]$ (*Figure 5—figure supplement 1A*).

For HMMs and MMs, first, state or mode sequences were drawn according to model transition probabilities, and then submode emissions were drawn according to model emission probabilities, conditioned on the state sequences sampled, to generate submode sequences (*Figure 5—figure supplement 1A,C,E*).

Exact matches were then counted between these synthetic sets and the first set of real sequences (*Equation 9*), and matched fraction reported (*Equation 10*, with N = 10).

## Stationarity

Stationarity of submode and mode frequencies over the experimental time period was confirmed by checking their distributions in data subsets over time, and by comparing empirical distributions with equilibrium distributions estimated from $\mathbf{T_{ij}}$ and $\mathbf{E}$.

## Dwell time analysis

Dwell times were calculated as follows. Trajectories were segmented according to Markov model $MM5$ mode definitions, producing mode sequences of length $k$, $\{(m_i)_k\}$, $m \in \{I \ldots V\}$, $i = 1 \ldots k$, and time sequences of $v_R$ extrema associated with each mode $m_i$, $\{(t_i)_k\}$. Only sequences with an observed switch into and out of Mode $m$ contributed to the dwell time distribution for Mode $m$. Hence, only sequences of length $k > 2$ were retained. Dwell time in Mode $m$ was taken as the difference between times of last and first sequential $v_R$ extrema classified as Mode $m$. Dwell times were also calculated by taking half the time to the $v_R$ extremum of last Mode $m_i$ as time of entry into mode $m$, and half the time to the $v_R$ extremum of the next Mode $m_j$ as time of exit from $m$, $m_i, m_j \neq m$. This extended dwell time calculation did not affect distribution shape (data not

shown). Dwell times in model-generated mode sequences were estimated the same way as in real sequences. The time base of all model-generated sequences was fixed such that the interval between modes was taken as a constant 167 ms. This time corresponds to the modal value of times between $v_R$ extrema.

## Test of velocity profile changes between training and test conditions

Velocity distributions $\{\Pr[\mathbf{v}]^{(m)} = \Pr[v_T, v_R, v_S \mid t']^{(m)}\}, m \in \{I \ldots V\}$, represent time-dependent velocity profiles for each mode $m$ specified by $MM5$. These distributions were compared between modes classified from Dataset S and Dataset L over $|t'| < 300$ ms. Distribution overlap was computed for each comparison using the Bhattacharya Coefficient for discrete distributions: $B(P, P') = \sum_{\mathbf{v}} [P(\mathbf{v})P'(\mathbf{v})]^{\frac{1}{2}}$ (*Kailath, 1967*). Significance was estimated by bootstrap (*Rice, 1994*): trajectory fragments representing the same transition type (same from: and to: states) were pooled from the two datasets, and resampled to generate 10000 comparisons between two randomly drawn trajectory subsets, maintaining the same sample sizes as in original sets. Each comparison produced an overlap value, and distributions of overlap values $\Pr[B]$ were used to evaluate the probability that the observed differences between two conditions are due to random chance while sampling from identical $\Pr[\mathbf{v}]^{(m)}$ distributions.

## Match test

Sequence subsets were sampled from a set of real behavior sequences or produced by a generative model. Subsets $R$ and $R'$ were drawn independently from a set of real sequences, and subset $S$ was drawn from a set of synthetic sequences. Different $R, R',$ and $S$ were drawn and compared $N$ times, as described for each analysis.

Some sequences may occur more than once in the same subset. Therefore, we counted sequence matches between two sets taking into account sequence multiplicity. For each unique element $x$ of multiset X, multiplicity $\nu_X(x)$ is the number of times element $x$ occurs in X. For two multisets X and Y, $\nu_{X \cap Y}(x) := min(\nu_X(x), \nu_Y(x))$.

For each unique sequence $r$ in subset $R$ and each unique sequence $s$ in subset $S$, with subsets of equal size $|R| = |S|$, define:

$$F^{R:R} := \frac{1}{|R|} \sum_r \nu_{R \cap R'}(r) \tag{8}$$

$$F^{S:R} := \frac{1}{|R|} \sum_s \nu_{S \cap R}(s) \tag{9}$$

$$f := \frac{\langle F^{S:R} \rangle_N}{\langle F^{R:R} \rangle_N} \tag{10}$$

where $\langle \cdot \rangle_N$ is an average over $N$ comparisons. Metric $f$ reports the average fraction of synthetic sequences found in a random real set, normalized to the average fraction of real sequences found in two randomly sampled real sets.

## Match test as a function of time

Mode sequences of length $k$, $(m)_k$ were sampled with replacement from sets of real or synthetic sequences. Real sequences were derived from Dataset L trajectories using the 15-submode segmentation criteria and 5-mode grouping criteria of $MM5$. Synthetic sequences were generated by iteratively sampling from state transition probability distributions of model $MM5$. The total number of sequences in each set ranged from 88434 ($k = 2$), to 5159 ($k = 20$). For each set of length $k$ sequences, $f_k$ was computed for subsets $\{(m_i)_{i=0}^k\}, k = 2 \ldots 20,$ with $|R| = |S| = 500, N = 5000.$ $\langle F^{R:R} \rangle_N$ ranged from 465 ($k=2$) to 96 ($k=20$). $\langle F^{S:R} \rangle_N$ ranged from 462 ($k=2$) to 76 ($k=20$).

**Table 4.** Materials and methods – Supplementary table 1. Comparison of dataset parameters, measurements, and findings.

| Work | Fly strain | Gender | Age (days, p. e.) | Recording Duration(min) | Fly speed (median, cm/sec) [c] | Measure-ment | Segmented behaviors, Total | Segmented behaviors, Locomotor |
|---|---|---|---|---|---|---|---|---|
| Berman et al. [a] | Oregon-R | ♂, ♀ | 1-14 | 60 | 0.3 (♂) | 2-D image (posture) | 117 | 17 |
| Berman et al. [b] | . | ♂ | . | . | . | . | . | . |
| Katsov et al. | Oregon-R | ♀ | 2-3 | 10 | 2.0 | body velocity | 15 | 15[d] |

[ . ] same value as row above.

[a] (**Berman et al., 2014**)

[b] (**Berman et al., 2016**)

[c] **Berman et al. (2014)** show median velocity ~0.3 mm/sec (Fig S1); the units appear to be a typographical error.

[d] Count includes submode 8/mode I, a mixture of locomotor and non-locomotor states (stops).

## Dataset and code availability

Full trajectory datasets and segmentation code are available at the Dryad Digital Repository (**Katsov et al., 2017**). The datasets include raw trajectories and annotation of submodes and modes.

## Summary of dataset and analysis parameters

Materials and methods - *Table 4*.

## Acknowledgements

The authors thank Daniel Ramot, Eran Mukamel, James Fitzgerald, and members of the Clandinin and Bargmann Labs for discussions, and Zak Frentz for comments on the manuscript.

## Additional information

### Funding

| Funder | Grant reference number | Author |
|---|---|---|
| Stanford University | Stanford Graduate Fellowship | Alexander Y Katsov Limor Freifeld |
| Jane Coffin Childs Memorial Fund for Medical Research | | Alexander Y Katsov |
| NIH Office of the Director | DP1 OD003530 | Thomas R Clandinin |
| National Institutes of Health | R01 EY022638 | Thomas R Clandinin |

The funders had no role in study design, data collection and interpretation, or the decision to submit the work for publication.

### Author contributions

AYK, Conceptualization, Data curation, Software, Formal analysis, Supervision, Validation, Investigation, Visualization, Methodology, Writing—original draft, Writing—review and editing, AK conceived the project and acquired data. AK, LF, MH, SK, and TC designed analyses. LF and AK performed Markov Model analysis, and AK performed all other analyses. AK, LF, SK and TC wrote the paper; LF, Conceptualization, Software, Formal analysis, Methodology, Writing—original draft, Writing—review and editing; MH, Supervision, Funding acquisition, Methodology; SK, Validation, Methodology, Writing—review and editing; TRC, Conceptualization, Formal analysis, Supervision, Funding acquisition, Validation, Methodology, Writing—original draft, Project administration, Writing—review and editing

## Author ORCIDs

Alexander Y Katsov, http://orcid.org/0000-0003-2155-3790
Seppe Kuehn, http://orcid.org/0000-0002-4130-6845
Thomas R Clandinin, http://orcid.org/0000-0001-6277-6849

## Additional files

### Major datasets

The following dataset was generated:

| Author(s) | Year | Dataset title | Dataset URL | Database, license, and accessibility information |
|---|---|---|---|---|
| Katsov AY, Freifeld L, Horowitz M, Kuehn S, Clandinin TR | 2017 | Data from: Dynamic structure of locomotor behavior in walking fruit flies | http://dx.doi.org/10.5061/dryad.854j2 | Available at Dryad Digital Repository under a CC0 Public Domain Dedication |

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
