## [Decision Letter]

Thank you for sending your work entitled "Dynamic structure of locomotor behavior in walking fruit flies" for consideration at *eLife*. Your article has been favorably evaluated by a Senior editor and three reviewers, one of whom, Ronald L Calabrese, is a member of our Board of Reviewing Editors.

The Reviewing editor and the other reviewer(s) discussed their comments before we reached this decision, and the Reviewing editor has assembled the following comments to help you prepare a revised submission.

In this manuscript the authors analyze an extensive data set of video recording of flies (*Drosophila melanogaster*) walking on flat surfaces. They analyze movements in 2 dimensions and ascertain three velocity components, translational, rotational, and side-slip. These data are then segmented and compared to a Markov model. They show that walking behavior exhibits two time scales, allowing movement dynamics to be segmented into distinct features. Bounds on velocity divergence define these elements. They conclude that over seconds, walking is a stochastic process composed of these elementary features. Within-feature divergence suggests that finite memory in stochastic behavioral sequences limits the effects of any one episode on subsequent behavior. The findings and methods are not only relevant to this important model system, but extend to other systems where analyses where a large number of behavioral sequences is recorded under normative and experimentally manipulated conditions.

Concerns:

The work is highly theoretical and very rigorous necessitating a high level of mathematical and theoretical by-in by the reader. There is a consensus that the authors should work to increase readability. All of the reviews taken together suggest that some rewriting will enhance the manuscript, and there are sufficient concrete suggestions in the original comments to guide such a revision.

Regarding finding "humanly recognizable" behavioral segments, as suggested by the first reviewer, it would be great if the authors could find a suitable example. On the other hand, if there are no such obvious examples of (or sequences of) modes that would have been picked up by visual inspection that in itself would be a significant finding worthy of mention, and we do not require an example. It is more important to tie sequences to particular computational principles.

There are several substantive primary and minor concerns in the reviews that the authors should address. Moreover, it seems important to address more squarely in the Discussion the issue of whether movement sequences are organized from primitive elements that act as attractors in the movement space.

Other issues to address:

Reviewer #1:

In this manuscript the authors analyze an extensive data set of video recording of flies (*Drosophila melanogaster*) walking on flat surfaces. They analyze movements in 2 dimensions and ascertain three velocity components, translational, rotational, and side-slip. These data are then segmented and compared to a Markov model. They show that walking behavior exhibits two time scales, allowing movement dynamics to be segmented into distinct features. Bounds on velocity divergence define these elements. They conclude that over seconds, walking is a stochastic process composed of these elementary features. Within-feature divergence suggests that finite memory in stochastic behavioral sequences limits the effects of any one episode on subsequent behavior.

Concerns:

The work is highly theoretical and very rigorous necessitating a high level of mathematical and theoretical by-in by the reader. This in itself is not bad but the authors make the paper needlessly inaccessible by failing to return to behavioral descriptions that are easily recognized and functionally meaningful. Only in paragraph four of subsection “Dynamic modes represent related submode groups, and extend behavior prediction” are we given any inkling of what humanly recognizable behavior corresponds to a mode and we are never given an example of a mode sequence in behavioral terms. The discussing is a letdown because, while it claims "Practically, the discovery that this compact model can largely recapitulate the behavioral repertoire of walking flies using descriptions of movement that operate on separate time scales will enable future investigation of how behavior on each time scale interfaces with sensory stimuli and neural control." The authors do not specify how this might be done. This is beautiful analysis of behavior but it is so abstract that the general reader will not be able to take away the message about how behavior is organized as suggested by the analysis.

A major rewrite is needed to make the work more accessible to a general readership by tying the analysis step-by-step to recognizable behaviors and behavior sequences. I realize that part of the message is that there are dynamical structures in the behavior that do not map directly onto humanly described behaviors but the authors must try as much as possible to make the work more real to biologists. There is much to admire in this study but it needs to be made much more accessible for an *eLife* audience; on the other hand *eLife* should support good theory and analysis like this work.

Reviewer #2:

The authors present a statistical analysis of behavioral patterns in walking fruit flies. Using an ICA-based analysis they identify major modes which, when viewed as states in a Markovian model, describe walking dynamics well on the 10s ms – few sec timescales. The work leads to several interesting insights that are elegantly discussed in the manuscript. The findings are interesting and their implications to future studies of neural control of behavior are expected to be significant.

Substantive concerns:

1) Subsection “Unbiased segmentation of trajectories defines modal movement patterns”: the text is attempting to describe in words the construction of the dataset for the ICA analysis without explicitly using mathematical definitions. The full description is provided in the methods section, so this comment concerns style and readability and the authors may choose to keep the text as is. I found the necessity to jump back and forth between the methods, the text, and Figure 3 to detract from the readability of this section. In my opinion, some mathematical definitions should appear in the main text and the caption of Figure 3 should be more detailed (e.g., a definition of the sign/constant, "c", is only given in the methods while it appears in the figure).

2) The authors should consider adding a page/box with the pseudocode of their various analysis steps. Having a compact, high-level, description of the logic guiding the analysis would be very convenient.

3) Making the analysis code available to researchers in the field would be useful (the comment on the availability of the datasets did not mention the analysis code).

Reviewer #3:

In the manuscript "Dynamic structure of locomotor behavior in walking fruit flies" the authors use the centroid and body-orientation velocity from richly sampled, two-dimensional trajectories of walking flies to quantitatively characterize fly movement. To the velocity time series, they apply an arsenal of computational techniques, including Principle Components Analysis, Independent Components Analysis, k-means clustering, hidden Markov modeling (HMM) and Markov modeling to argue that walking velocity dynamics can be broadly divided into two components; on short timescales (~100 ms), trajectories consist of relatively stereotyped bouts ultimately characterized as 5 discrete modes (Figure 6) while on longer timescales (~1s, about the correlation time of the velocity auto-correlation functions) the dynamics consists of transitions between modes. In support of this two-component perspective they show that the distribution of dwell times within each behavioral mode is approximately exponential with transitions to the next state mainly dependent on the current and previous state (a Markov model of order 1).

Overall, I found the manuscript and research very engaging and the problem of quantitively characterizing behavior is both topical and fundamental. However, while I believe that this work will ultimately add significantly to our understanding of natural behavior and it's analysis, the presentation needs to be substantially streamlined so as to be understandable to the widest possible audience. Perhaps most importantly is an issue of clarity. The manuscript is currently written as if to follow the analysis very much as it happens. While this can be pleasant and useful for an expert, I worry that most of the intended audience will lose track of the fundamental points. For example, the title of the first section, "Behavior episodes are defined by bounded divergence", is complex and confusing. Within this section is first a finding that the "phase space" pair of the instantaneous velocity and acceleration (v,a) is not, by itself, a useful candidate for behavioral segmentation as the same pair is connected to multiple movement patterns. Thus the transition to looking at behavioral time traces centered on peaks in v_r. Why not just introduce the segmentation directly and then move the previous motivation to supplementary material where it won't distract from the main thrust? Indeed, I had the same feeling in the sections introducing the submode and mode sequences. There is a fundamental and primary computation here, which is how to characterize the space of velocity trajectories, sampled from short windows around peaks in v_r. But with modes and submodes and HMM's and MM's both the terminology and the story get quickly complicated.

Finally, throughout the manuscript I feel that the authors orbit obliquely around a general and fundamental question: what is the origin of complex, behavioral dynamics. In particular, and perhaps with human language as a metaphor, can complex behaviors be built from more primitive parts and, if so, what might those parts look like? Are these primitive elements attractors in the movement space? This is an appealing idea with some support in the literature. But in this system and analysis the authors argue "no" as attractors are uncommon in higher-dimensional spaces and their analysis suggests that trajectories represented by the primitives are unstable so that differences grow with time over the duration of the primitive. Instead, movements are "stabilized" by transitioning between primitives. If I've gotten that summary correct, then it needs to more clearly written and developed in the text.

[Editors' note: further revisions were requested prior to acceptance, as described below.]

Thank you for submitting your article "Dynamic structure of locomotor behavior in walking fruit flies" for consideration by *eLife*. Your article has been reviewed by three peer reviewers, one of whom, Ronald L Calabrese (Reviewer #1), is a member of our Board of Reviewing Editors and the evaluation has been overseen by Eve Marder as the Senior Editor. The following individuals involved in review of your submission have agreed to reveal their identity: David Biron (Reviewer #2).

The reviewers have discussed the reviews with one another and the Reviewing Editor has drafted this decision to help you prepare a revised submission.

Summary:

This is an interesting manuscript that describes a new unbiased method for behavioral segmentation of spontaneous locomotory behavior in fruit flies. After analyzing video images of ~1,000,000 behavioral trajectories in velocity phase space, the authors take time points before and after peaks in rotational velocity to identify movement trajectories. Because the dynamics of spontaneous locomotion are consistent over hundreds of milliseconds, elementary features can be defined. They align the trajectories by their rotational velocity peaks and use ICA to identify independent movement submodes, only some of which correspond to distinct behaviors recognized by human observers or machine classification. This underscores that behavior dynamics may be organized differently relative to behavior features salient to human observers, or machine classification that collapses time scales.

The submodes are then linked into modes by likelihood of sequential occurrence. A Markov model with five states is then generated that predicts movement sequences and where the five states correspond to modes comprising multiple submodes. Using this Markov model they then show that spontaneous locomotory behavior is well described by a stochastic process composed of these 5 modes. Within submodes, velocities diverge, suggesting that dynamical stability of movement submodes is a weak behavioral constraint. Rather, the effects of any one submode on subsequent behavior are limited by finite memory in a stochastic behavior sequence.

The paper is well written and the illustrations get the main data across adequately. All necessary data are presented. The experiments appear carefully analyzed and appropriate statistical methods are used. The modeling appears rigorous. The findings are novel and should be of interest to those studying how elemental motor behaviors are organized into overt behavior and behavioral sequences or who are interested in automatic methods for detecting elemental behaviors and their dynamic organization.

Essential revisions:

This submission is a bit unusual as it is a revised submission after a very prolonged hiatus. Thus this review is somewhat of a hybrid for a new submission and a revision.

The expert reviewers were in substantial agreement so that their major comments are appended to assist in revision. The authors should focus their revision on three major points.

1) The computer codes for all the modeling and analyses with appropriate documentation must be made available. *eLife* will assist in this process and makes Dryad available to authors.

2) A stronger attempt should be made to compare the work here to that of Berman et al., 2014, 2016 and to make explicit the implications of the difference in approach (posture vs. velocity). The expert reviews detail how this might be accomplished.

3) There is still some concern that the authors could do more to relate the submodes/modes they observe to humanly recognizable behaviors. Because it may be difficult to make convincing argument for biological relevance of behavioral features not salient to human observers and further there is no guarantee that the modes identified here will remain under different conditions (although more general features may well), the authors should also focus more on the issue of dynamic stability. Perhaps this does not come through strongly enough.

Reviewer #2:

This is a revised version of a manuscript originally submitted in 2014. The manuscript was originally favorably reviewed and it is still relevant, novel, rigorous, and engaging. The authors have responded comprehensively to the original review comments and the accessibility to a broader readership was improved.

As noted, Berman et al., 2016 is now published. The two manuscripts are complementary and may well become foundational. As things stand, fully appreciating the similarities and differences between them would require reading them side by side. The underlying reasons for the differences between the two can be subtle. Contrary to common instinct, these so-called 'discrepancies' should be put front and center. They are informative and relevant to pertinent questions in the field.

The authors' rebuttal contains a long discussion of these points. Disappointingly, the manuscript itself only hints at them in the 'Timescales in behavior" Discussion section. In my opinion, the Discussion section should include a concise version of the 'Additional note' from the rebuttal. It would detail the comparison between these two sets of results and explicitly explain hypothesized reasons for differences.

Moreover, I would advocate a Table (or box) summarizing these points and their implications. Such a table would span the differences in experimental conditions that may affect physiology and experimental resolution, summarize implications of choosing posture vs. velocity, list hypothesized distinctions between locomotor and non-locomotor primitives, etc. It would be broadly useful to gain some appreciation of these subtleties and considerations at a glance.

Reviewer #3:

On the positive side:

This paper uses complex modeling techniques to segment locomotor behavior in a principled, unbiased manner.

It therefore has the potential to provide a useful approach for how the field analyzes and thinks about behavioral sequences in an unbiased way.

From previous work, there are a number of ways to segment and analyze *Drosophila* locomotion data, however, most depended on human-set definitions of behaviors. The most notable exception is the work from Berman et al., (2014 and 2016) in which behaviors were also characterized de novo. Of potential interest to experts in the field, the details and even some conclusions differ with the present study.

One potential advantage of this and other unbiased approaches is that it uncovers relations between behaviors that are not readily obvious to human observation.

Room for improvement:

Although equations are provided, the computer code is not (at least that I could find). This would need to be provided prior to publication, with clear documentation.

The ethological relevance of these behavioral submodes, modes, and sequences is not clear. The paper would be significantly more influential if a case could be made for why these modes help biologists understand specific behaviors. Currently, one is left with a very principled analysis that seems not well-grounded in biological reality.

A more direct comparison with previous data (e.g., the original ctrax paper comparing males and females) may help breathe some biological context into the modes and submodes defined here.

Finally, without a clearer biological application, the authors need to better justify why this approach is better than previous unbiased analyses of behavior. Although differences with Berman et al. are discussed, it is not clear if this approach is an improvement, or just a valid alternative.

---

## [Author Response]

*Concerns:*

The work is highly theoretical and very rigorous necessitating a high level of mathematical and theoretical by-in by the reader. There is a consensus that the authors should work to increase readability. All of the reviews taken together suggest that some rewriting will enhance the manuscript, and there are sufficient concrete suggestions in the original comments to guide such a revision.

The manuscript has been revised from beginning to end. Figures have been improved, reorganized, and new figures and explanatory boxes were added. We believe that we have addressed every reviewer suggestion, and are grateful for the constructive feedback, which has significantly improved the manuscript.

Regarding finding "humanly recognizable" behavioral segments, as suggested by the first reviewer, it would be great if the authors could find a suitable example. On the other hand, if there are no such obvious examples of (or sequences of) modes that would have been picked up by visual inspection that in itself would be a significant finding worthy of mention, and we do not require an example. It is more important to tie sequences to particular computational principles.

We've added comparisons between this work's segmentation and humanly recognizable behaviors, in both the Results and Discussion sections. We've also revised Figure 4 to illustrate humanly recognizable behaviors in individual modes and added two additional figures: Figure 3—figure supplement 2 to show samples of trajectories from each submode class, and Figure 5—figure supplement 1) to show samples of mode sequences.

There are several substantive primary and minor concerns in the reviews that the authors should address. Moreover, it seems important to address more squarely in the Discussion the issue of whether movement sequences are organized from primitive elements that act as attractors in the movement space.

A Discussion section has been added ("Mode structure and stability").

Other issues to address:

Reviewer #1:

In this manuscript the authors analyze an extensive data set of video recording of flies (Drosophila melanogaster) walking on flat surfaces. They analyze movements in 2 dimensions and ascertain three velocity components, translational, rotational, and side-slip. These data are then segmented and compared to a Markov model. They show that walking behavior exhibits two time scales, allowing movement dynamics to be segmented into distinct features. Bounds on velocity divergence define these elements. They conclude that over seconds, walking is a stochastic process composed of these elementary features. Within-feature divergence suggests that finite memory in stochastic behavioral sequences limits the effects of any one episode on subsequent behavior.

Concerns:

The work is highly theoretical and very rigorous necessitating a high level of mathematical and theoretical by-in by the reader. This in itself is not bad but the authors make the paper needlessly inaccessible by failing to return to behavioral descriptions that are easily recognized and functionally meaningful. Only in paragraph four of subsection “Dynamic modes represent related submode groups, and extend behavior prediction” are we given any inkling of what humanly recognizable behavior corresponds to a mode and we are never given an example of a mode sequence in behavioral terms.

We have now tied in behavior descriptions throughout the revised manuscript, and have provided examples of mode sequences

The discussing is a letdown because, while it claims "Practically, the discovery that this compact model can largely recapitulate the behavioral repertoire of walking flies using descriptions of movement that operate on separate time scales will enable future investigation of how behavior on each time scale interfaces with sensory stimuli and neural control." The authors do not specify how this might be done. This is beautiful analysis of behavior but it is so abstract that the general reader will not be able to take away the message about how behavior is organized as suggested by the analysis.

The discussion has been extensively revised, adding concrete interpretations of our work throughout. Discussion of the interface between sensory stimuli and behavior structure was removed to more tightly focus discussion on the structure of behavior.

A major rewrite is needed to make the work more accessible to a general readership by tying the analysis step-by-step to recognizable behaviors and behavior sequences. I realize that part of the message is that there are dynamical structures in the behavior that do not map directly onto humanly described behaviors but the authors must try as much as possible to make the work more real to biologists. There is much to admire in this study but it needs to be made much more accessible for an eLife audience; on the other hand eLife should support good theory and analysis like this work.

The entire manuscript has been extensively re-written to improve accessibility.

Reviewer #2:

Substantive concerns:

1) Subsection “Unbiased segmentation of trajectories defines modal movement patterns”: the text is attempting to describe in words the construction of the dataset for the ICA analysis without explicitly using mathematical definitions. The full description is provided in the methods section, so this comment concerns style and readability and the authors may choose to keep the text as is. I found the necessity to jump back and forth between the methods, the text, and Figure 3 to detract from the readability of this section. In my opinion, some mathematical definitions should appear in the main text and the caption of Figure 3 should be more detailed (e.g., a definition of the sign/constant, "c", is only given in the methods while it appears in the figure).

The legend of Figure 3 has been expanded, and a new [Box B1] was added to consolidate a number of mathematical definitions inline with the main text.

2) The authors should consider adding a page/box with the pseudocode of their various analysis steps. Having a compact, high-level, description of the logic guiding the analysis would be very convenient.

A pseudocode box ([Box B2]) was added to describe all analysis steps.

3) Making the analysis code available to researchers in the field would be useful (the comment on the availability of the datasets did not mention the analysis code).

Analysis code is made available along with the datasets.

Reviewer #3:

In the manuscript "Dynamic structure of locomotor behavior in walking fruit flies" the authors use the centroid and body-orientation velocity from richly sampled, two-dimensional trajectories of walking flies to quantitatively characterize fly movement. To the velocity time series, they apply an arsenal of computational techniques, including Principle Components Analysis, Independent Components Analysis, k-means clustering, hidden Markov modeling (HMM) and Markov modeling to argue that walking velocity dynamics can be broadly divided into two components; on short timescales (~100 ms), trajectories consist of relatively stereotyped bouts ultimately characterized as 5 discrete modes (Figure 6) while on longer timescales (~1s, about the correlation time of the velocity auto-correlation functions) the dynamics consists of transitions between modes. In support of this two-component perspective they show that the distribution of dwell times within each behavioral mode is approximately exponential with transitions to the next state mainly dependent on the current and previous state (a Markov model of order 1).

Overall, I found the manuscript and research very engaging and the problem of quantitively characterizing behavior is both topical and fundamental. However, while I believe that this work will ultimately add significantly to our understanding of natural behavior and it's analysis, the presentation needs to be substantially streamlined so as to be understandable to the widest possible audience. Perhaps most importantly is an issue of clarity. The manuscript is currently written as if to follow the analysis very much as it happens. While this can be pleasant and useful for an expert, I worry that most of the intended audience will lose track of the fundamental points. For example, the title of the first section, "Behavior episodes are defined by bounded divergence", is complex and confusing. Within this section is first a finding that the "phase space" pair of the instantaneous velocity and acceleration (v,a) is not, by itself, a useful candidate for behavioral segmentation as the same pair is connected to multiple movement patterns. Thus the transition to looking at behavioral time traces centered on peaks in v_r. Why not just introduce the segmentation directly and then move the previous motivation to supplementary material where it won't distract from the main thrust? Indeed, I had the same feeling in the sections introducing the submode and mode sequences. There is a fundamental and primary computation here, which is how to characterize the space of velocity trajectories, sampled from short windows around peaks in v_r. But with modes and submodes and HMM's and MM's both the terminology and the story get quickly complicated.

These points are both valid, and valuable. We have made a lot of effort to streamline the manuscript, and we hope the manuscript is now better integrated.

Specifically, the first Results section was revised to emphasize two important points: (1) how time scales underlying the rest of the work were arrived at in a model-independent manner, and (14) how behavior episodes, to which different models were applied in subsequent sections, were defined from an initial, model-independent analysis.

The section on submodes was revised to emphasize some properties of behaviour uncovered in the first stage of analysis. While subsequent analysis showed that different submodes can be statistically interchangeable within a mode, submodes are reported as a potentially distinct level of behavior organization.

The section on Markov models was revised to clarify both motivation and results. Analysis started from Hidden Markov models as a more general approach than Markov models. After obtaining results from HMM analysis, the more restrictive assumptions of Markov models were better justified for subsequent work. The text was streamlined to retain this basic point.

Finally, throughout the manuscript I feel that the authors orbit obliquely around a general and fundamental question: what is the origin of complex, behavioral dynamics. In particular, and perhaps with human language as a metaphor, can complex behaviors be built from more primitive parts and, if so, what might those parts look like? Are these primitive elements attractors in the movement space? This is an appealing idea with some support in the literature. But in this system and analysis the authors argue "no" as attractors are uncommon in higher-dimensional spaces and their analysis suggests that trajectories represented by the primitives are unstable so that differences grow with time over the duration of the primitive. Instead, movements are "stabilized" by transitioning between primitives. If I've gotten that summary correct, then it needs to more clearly written and developed in the text.

At the reviewer's suggestion, we developed discussion on the nature of primitive parts in behavior, and specifically the extent to which behavior elements can be described as attractors in movement space. Additional analysis was also performed to explore this point more concretely.

[Editors' note: further revisions were requested prior to acceptance, as described below.]

Essential revisions:

*1) The computer codes for all the modeling and analyses with appropriate documentation must be made available. eLife will assist in this process and makes Dryad available to authors.*

Computer code has been deposited at Dryad, along with documentation, code parameter files, and datasets.

*2) A stronger attempt should be made to compare the work here to that of Berman et al., 2014, 2016 and to make explicit the implications of the difference in approach (posture vs. velocity). The expert reviews detail how this might be accomplished.*

Two Discussion sections have been added to compare this work with Berman et al., 2014, 2016, and to make explicit the implications of different approaches to behavior segmentation.

3) There is still some concern that the authors could do more to relate the submodes/modes they observe to humanly recognizable behaviors. Because it may be difficult to make convincing argument for biological relevance of behavioral features not salient to human observers and further there is no guarantee that the modes identified here will remain under different conditions (although more general features may well), the authors should also focus more on the issue of dynamic stability. Perhaps this does not come through strongly enough.

Several sections throughout the manuscript have been re-written to emphasize biological relevance and issues pertaining to dynamic stability (Abstract, Introduction, Discussion). Also, to help relate segmented behaviors to humanly recognizable movements, illustrations have been added as suggested by reviewer 1.

*Reviewer #2:*

*This is a revised version of a manuscript originally submitted in 2014. The manuscript was originally favorably reviewed and it is still relevant, novel, rigorous, and engaging. The authors have responded comprehensively to the original review comments and the accessibility to a broader readership was improved.*

*As noted, Berman et al., 2016 is now published. The two manuscripts are complementary and may well become foundational. As things stand, fully appreciating the similarities and differences between them would require reading them side by side. The underlying reasons for the differences between the two can be subtle. Contrary to common instinct, these so-called 'discrepancies' should be put front and center. They are informative and relevant to pertinent questions in the field.*

*The authors' rebuttal contains a long discussion of these points. Disappointingly, the manuscript itself only hints at them in the 'Timescales in behavior" Discussion section. In my opinion, the Discussion section should include a concise version of the 'Additional note' from the rebuttal. It would detail the comparison between these two sets of results and explicitly explain hypothesized reasons for differences.*

Two sections have been added to consider the implications of measurement type, experimental conditions, and other variables in studies of the structure of behavior dynamics. Each section contains concise comparisons with Berman et al., hypotheses about differences, and issues that can be clarified with further work.

*Moreover, I would advocate a Table (or box) summarizing these points and their implications. Such a table would span the differences in experimental conditions that may affect physiology and experimental resolution, summarize implications of choosing posture vs. velocity, list hypothesized distinctions between locomotor and non-locomotor primitives, etc. It would be broadly useful to gain some appreciation of these subtleties and considerations at a glance.*

A table has been constructed listing specific differences between studies, but no simple way was found to present hypotheses in a table format. It is hoped that the discussion of differences is concise enough in the main text.

*Reviewer #3:*

*Room for improvement:*

*Although equations are provided, the computer code is not (at least that I could find). This would need to be provided prior to publication, with clear documentation.*

Code with documentation is available at Dryad (DOI: http://dx.doi.org/10.5061/dryad.854j2). At the time of review, Dryad had two different submissions from us, one of which was associated with the previous (outdated) version of this manuscript. That version was missing code. It is now removed. We apologize for this.

Data and code are in the following categories:

1) Datasets containing all velocity trajectories and their annotation after segmentation

2) Matlab scripts and their parameter files, operating on velocity trajectories

a) to segment trajectories into submodes

b) to assign submodes to modes

3) Matlab scripts and their parameter files, operating on segmented trajectories

a) to perform match tests to compare real and synthetic trajectory distributions

b) to generate synthetic trajectories according to model

*The ethological relevance of these behavioral submodes, modes, and sequences is not clear. The paper would be significantly more influential if a case could be made for why these modes help biologists understand specific behaviors. Currently, one is left with a very principled analysis that seems not well-grounded in biological reality.*

We have re-written parts of the text to emphasize the aims of this work. Its ultimate aim is to improve understanding of the organization of behavior, and in so doing to help constrain investigations of the neural circuits that control it. To bridge understanding of behavior with studies of neural circuits, progress needs to be made in understanding behaviors at different time scales. This is the goal of analyses begun in this work. As behaviors are examined using different approaches, including ours, it is hoped that the biological significance of both specific behaviors, and their relationships as a repertoire will be clarified.

*A more direct comparison with previous data (e.g., the original ctrax paper comparing males and females) may help breathe some biological context into the modes and submodes defined here.*

We have expanded discussion of previous data, including the original ctrax paper.

*Finally, without a clearer biological application, the authors need to better justify why this approach is better than previous unbiased analyses of behavior. Although differences with Berman et al. are discussed, it is not clear if this approach is an improvement, or just a valid alternative.*

We have now expanded discussion on differences between our approaches to data collection and analysis, and prior work. Our work contributes large datasets acquired at stable experimental conditions in which flies remain active for the duration of the experiment, methods for direct analysis of behavior dynamics, and preliminary findings on the structure of behavior under these experimental conditions. Ultimately, we expect that analysis of behavior and its control will benefit from an aggregate of different methods.